# Consistent patterns of common species across tropical tree communities

Trees structure the Earth's most biodiverse ecosystem, tropical forests. The vast number of tree species presents a formidable challenge to understanding these forests, including their response to environmental change, as very little is known about most tropical tree species. A focus on the common species may circumvent this challenge. Here we investigate abundance patterns of common tree species using inventory data on 1,003,805 trees with trunk diameters of at least 10 cm across 1,568 locations[1–6] in closed-canopy, structurally intact old-growth tropical forests in Africa, Amazonia and Southeast Asia. We estimate that 2.2%, 2.2% and 2.3% of species comprise 50% of the tropical trees in these regions, respectively. Extrapolating across all closed-canopy tropical forests, we estimate that just 1,053 species comprise half of Earth's 800 billion tropical trees with trunk diameters of at least 10 cm. Despite differing biogeographic, climatic and anthropogenic histories[7], we find notably consistent patterns of common species and species abundance distributions across the continents. This suggests that fundamental mechanisms of tree community assembly may apply to all tropical forests. Resampling analyses show that the most common species are likely to belong to a manageable list of known species, enabling targeted efforts to understand their ecology. Although they do not detract from the importance of rare species, our results open new opportunities to understand the world's most diverse forests, including modelling their response to environmental change, by focusing on the common species that constitute the majority of their trees.

Tropical forests are a crucial component of the Earth system; they cover around 10% of the Earth's land surface[8] but contribute approximately 33% of terrestrial net primary productivity[9]. They account for around 40% of the carbon stored in live vegetation[10] and are globally important carbon sinks[11]. Tropical forests are also extraordinarily biodiverse, harbouring two-thirds of all known species[12] and the majority of the world's biodiversity hotspots[13]. Of note, as many tree species can be found in a single hectare of tropical forest as in the entire native Western European tree flora[14]. Recent estimates suggest that there are approximately 37,900 named tropical tree species in the scientific literature[15], with potentially thousands more yet to be identified by scientists[16]. This extraordinary diversity means that little is known about the biology of the vast majority of tropical tree species. Our understanding of tropical forest ecology, productivity and carbon storage and how they may respond to environmental change is hindered by this lack of knowledge. This limited understanding also curtails scientific input into land use, biodiversity, climate and other forest-related policy and management.

Our understanding of tropical forests may improve through a focus on the most common tree species. This is a promising avenue, given that species abundance distributions (SADs) showing a modest number of common species and much larger numbers of rare species have been documented across taxa globally[17–19]. Indeed, analyses of tropical forest inventory data from Amazonia have shown that a relatively small number of common species comprise a majority of trees in the region[6,20–24]. However, whether such patterns hold in other tropical forests is unknown, as there have been no comparable analyses for African or Southeast Asian tropical forests. Perhaps, given the substantial differences in total tree species richness[25], forest structure[1], contemporary climate[26] and biogeographic and human-occupancy histories[7] among continents, important contrasts in patterns of common species would be expected. Alternatively, if the same processes or mechanisms apply to all tropical forests[27], highly consistent patterns may be expected. Crucially, if a tractably modest number of common species do comprise the majority of tropical trees on Earth, this could open new ways of understanding tropical forests by investigating the ecology of the common species.

Cross-continental comparisons of common species patterns are complicated by unresolved differences in the results from published Amazon forest studies[6,20,22]. Estimates of hyperdominance—describing the minimum number of species required to account for 50% of all trees in a sample—range from 1.4% to 8.2% of the total number of species found in each of the Amazon forest datasets analysed (corresponding to 224 and 1,312 hyperdominant species respectively, assuming 16,000 Amazon tree species). Therefore, here we: (1) investigate sample-related biases and standardize our sampling to enable meaningful comparisons among datasets; (2) test whether patterns of hyperdominance differ across Amazonia, Africa and Southeast Asia; (3) extrapolate our results to assess how many species comprise half of all Earth's tropical trees; (4) assess species abundance patterns, with differing classifications of 'common species' beyond hyperdominance; and (5) use resampling techniques to assess which sampled species are likely to be hyperdominant.

We analyse species abundance data from networks of inventory plots across three continents. We limit our analysis to closed canopy

A list of authors and their affiliations appears online. ✉e-mail: declan.cooper.16@ucl.ac.uk; s.l.lewis@ucl.ac.uk

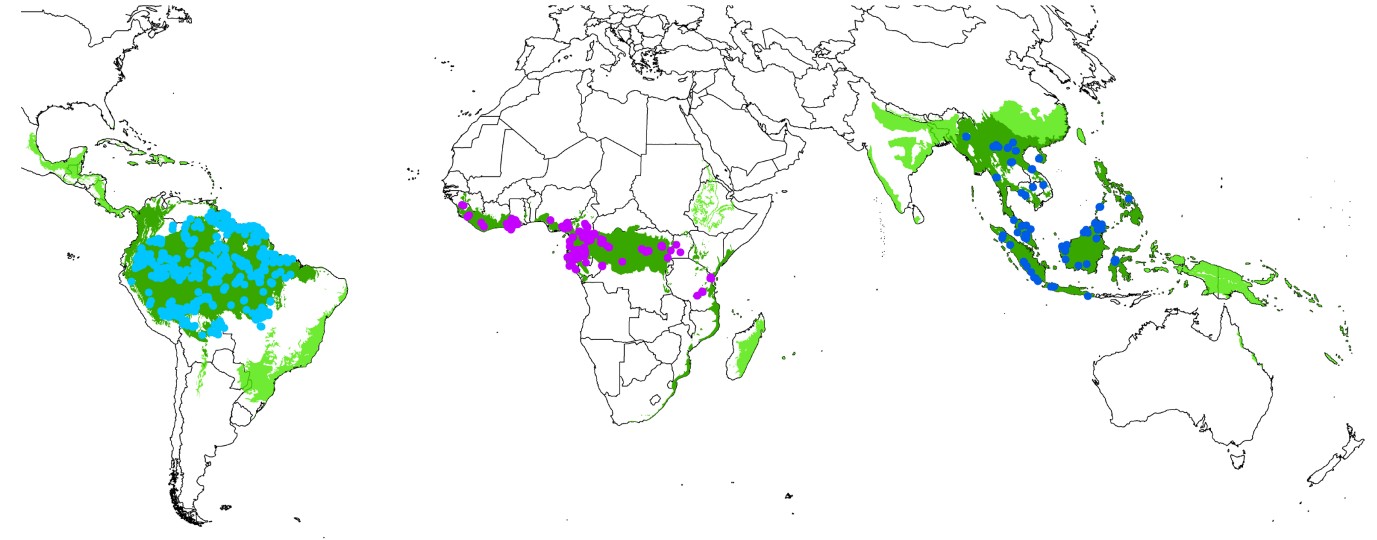

**Fig. 1 | Location of the 1,568 plots, tropical forest regions, and tropical forest biome extent used in the study.** Dots show the location of the plots analysed, coloured by continental region. Dark green shows the Amazonia, Africa and Southeast Asia regions that we extrapolate to. Light green shows 'tropical and subtropical moist broadleaf forests'[60], which we extrapolate to as the closed canopy tropical forest biome.

structurally intact old-growth tropical forests. For Amazonia, defined as the lowland Amazon Basin and Guiana Shield, we use the Amazon Tree Diversity Network and RAINFOR datasets ($n$ = 1,097 plots). For Africa, encompassing West, central and East Africa, we use the African Tropical Rainforest Observatory Network (AfriTRON)[1], Central African Plot Network, and two smaller networks[2,3] ($n$ = 368 plots). For Southeast Asia, defined as extending from Myanmar in the West to Sulawesi in the East, we use a tree diversity[4] and a carbon monitoring[5] network ($n$ = 103 plots). We limit our analysis to trees with trunk diameter of at least 10 cm at breast height (1.3 m along the stem or above any buttresses or deformities), the widely used minimum size for inventorying tropical trees. The combined dataset includes 1,003,805, trees, of which 93.3% are identified to species (Fig. 1 and Extended Data Table 1).

## Consistent patterns of commonness

The Africa, Amazonia and Southeast Asia datasets differ in the number and size of plots sampled and the number of trees sampled (Extended Data Table 1). We therefore excluded small plots (below 0.9 ha; Extended Data Fig. 1 and Methods) and used rarefaction—that is, repeated random subsampling of plots to comparable numbers of trees—to standardize sampling across the three datasets (Fig. 2).

Rarefying to a common sample size of 77,587 stems, the size of the Asia dataset (equivalent to 150, 116 and 103 plots in Africa, Amazonia and Southeast Asia respectively), we find that 77 species (95% confidence interval: 62–92) in Africa comprise 50% of individual trees, compared with 174 species (95% confidence interval: 134–215) in Amazonia and 172 species (95% confidence interval: 125–217) in Southeast Asia (Table 1 and Fig. 2). However, the substantially lower number of hyperdominant species in Africa compared with Amazonia and Southeast Asia scales with the substantially lower number of total species. We find just 1,132 species in our standardized 77,587 tree sample in Africa, compared with 2,565 and 2,585 species in Amazonia and Southeast Asia, respectively for the same sample size. Consequently, percentage hyperdominance is statistically indistinguishable among the continents at 6.79% (95% confidence interval: 5.39%–8.20%), 6.80% (95% confidence interval: 5.24%–8.36%) and 6.65% (95% confidence interval: 4.59%–8.71%) in Africa, Amazonia and Southeast Asia, respectively (Table 1). This consistency is not affected by the aggregated spatial distribution of plots within each region (Extended Data Fig. 2) and holds true for analyses based solely on 1-ha plots (Methods). Thus, once sampling is standardized, there is marked pan-tropical consistency in the proportion of the total number of tree species accounted for by the most common species.

The consistency of commonness is not limited to defining common species as those that account for 50% of all individual trees in a dataset. The proportions of the total number of species required to account for thresholds between 10% and 90% of individual trees are also highly consistent across the rarefied data for the three continents (Fig. 3 and Extended Data Table 3). Thus, the data from the three continents appear to result from the same underlying statistical distribution.

Our rarefaction analysis shows that the number of hyperdominants, the total number of species and the percentage hyperdominance are dependent on sample size. This is because as plots—and therefore trees—are added to the sample, increasing numbers of rare species start to appear. Meanwhile, most common species have, by definition, already appeared, but their abundances increase. Thus, with increasing sample size, the number of hyperdominants increases, but at an ever-decreasing rate that tends towards saturation (Fig. 2 and Extended Data Fig. 3). The total number of species increases at a decreasing rate with increasing sample size, without apparent saturation. Therefore, as sample sizes increase, the percentage hyperdominance decreases gradually, but does not appear to saturate (Fig. 2 and Extended Data Fig. 3). This sample size dependence is likely to explain the published differences in percentage hyperdominance in Amazonian forests, which follow expectations given the sample size in each study[6,20,22].

Amazonia and Southeast Asia show remarkably similar patterns of commonness and diversity. The rarefaction curves of the number of species accounting for 50% of all trees (Fig. 2a), total number of species (Fig. 2b), percentage hyperdominance (Fig. 2c) and Fisher's α—the parameter of the log series distribution shown to best describe tropical tree species abundance distributions[21] (Fig. 2d)—are almost identical between the two datasets. Furthermore, the numbers of species required to account for any threshold between 10% and 90% of trees in the respective rarefied samples of 77,587 trees are statistically indistinguishable (Table 1 and Extended Data Tables 2 and 3). This equivalence in overall tropical forest diversity patterns between these similarly species-rich regions is particularly striking given their very different biogeographic, climatic and anthropogenic histories, and the fact that Amazonia is one large contiguous region, whereas Southeast Asia is a series of islands and island-like regions.

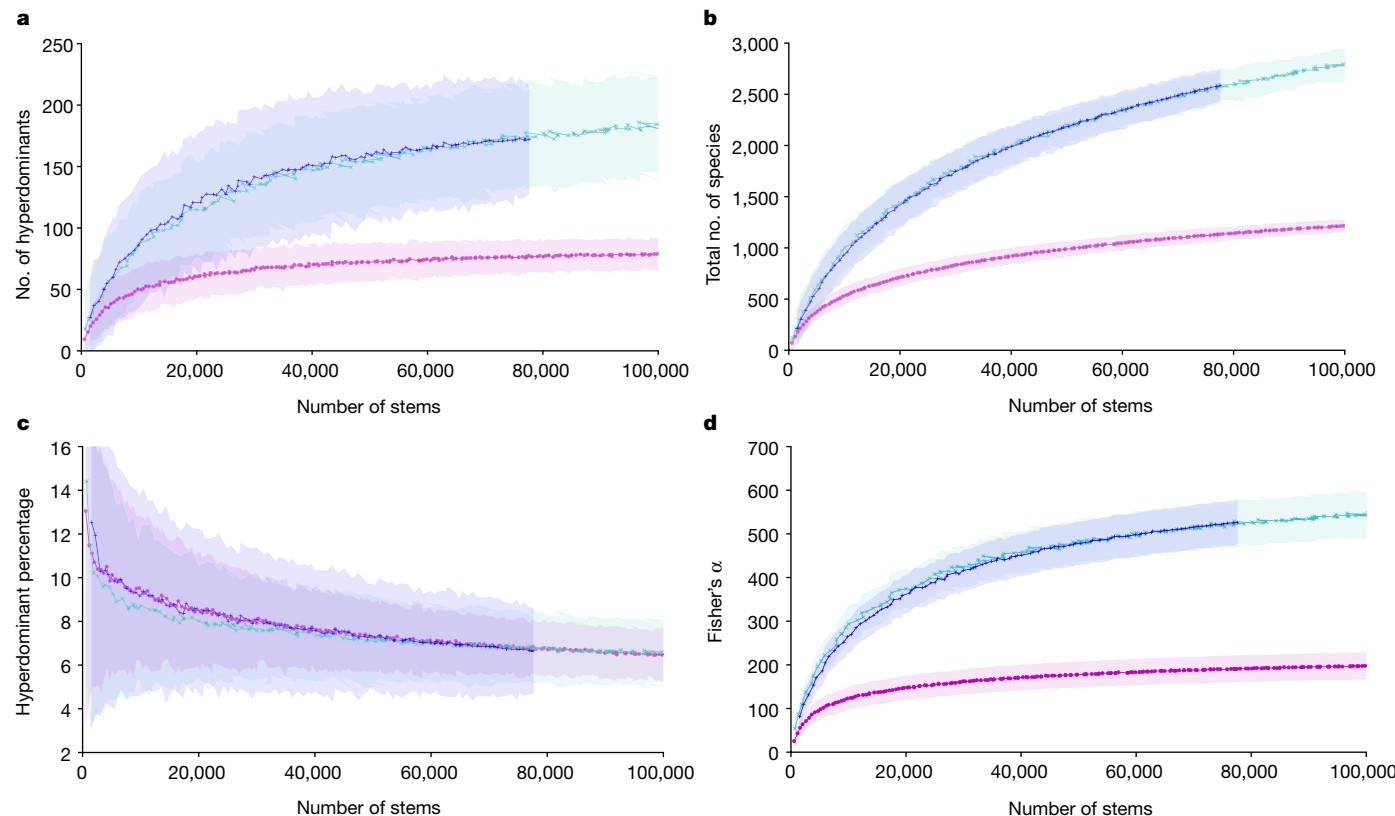

**Fig. 2 | Rarefaction curves showing the effect of increasing sample size on the number of hyperdominants, total species, hyperdominant percentage and fitted values of Fisher's α in tropical tree communities. a–d,** The effect of increasing sample size on the number of hyperdominants (**a**), total species (**b**), hyperdominant percentage (**c**) and fitted values of Fisher's α (**d**) in tropical Africa (magenta), Amazonia (cyan), Southeast Asia (blue). Rarefied data (mean values across iterations of subsamples) are shown as points joined by lines for clarity, shaded areas represent 95% confidence intervals (derived via the s.d. across iterations of subsamples taken with replacement at each sampling point). Note that resampling for rarefaction was by subsampling of plots, but curves are re-plotted on an *x* axis of number of stems.

In contrast to the similarity between Amazonia and Southeast Asia, our results provide sample size-corrected validation of the 'odd-one-out' observation[28,29] of much lower tree species richness in Africa compared with Amazonia and Southeast Asia. Here we add a similar odd-one-out observation of a much lower number of common species in Africa than in Amazonia and Southeast Asia. However, in combination these two results lead to an almost identical percentage hyperdominance in the African, Amazonian and Southeast Asian rarefied data. This consistency extends to the proportion of species required to account for all thresholds between 10% and 90% of trees in the rarefied data (Fig. 3 and Extended Data Table 3). This pan-tropical invariance recasts the tropical forests of Africa from 'odd' in terms of species richness to statistically indistinguishable from those in Amazonia and Southeast Asia in terms of proportional patterns of abundance. Overall, using standardization by rarefaction, we find consistent patterns of species abundance across Africa, Amazonia and Southeast Asia.

## Scaling to the study region

Next, we estimate commonness patterns in each of our three study regions: Africa, Amazonia and Southeast Asia. We extrapolate log series fits to the empirical Africa, Amazonia and Southeast Asia datasets (Extended Data Fig. 4), including a correction to account for the clumped spatial occurrence of species, to the total number of trees with trunk diameter of at least 10 cm in each study region. We estimate that just 104 species (95% confidence interval: 101–107) account for 50% of the 113 billion trees in Africa's closed canopy tropical forests (Table 2). We also estimate that just 299 species (95% confidence interval: 295–304) account for 50% of the 344 billion trees in Amazonia's closed canopy tropical forest, and 278 (95% confidence interval: 268–289) account for 50% of the 129 billion trees in Southeast Asia's closed canopy tropical forests (Table 2). Our results from Amazonia match those derived using a different extrapolation approach[30].

Our extrapolations again outline consistent percentage hyperdominance: just 2.2% of African, 2.2% Amazonian and 2.3% of Southeast Asian species account for 50% of all trees with trunk diameters of at least 10 cm in each region (Table 2). The dominant proportions of total species required to account for 10% to 90% of trees are also very similar across continents (Fig. 3 and Extended Data Table 5). The lower percentage dominance values from the extrapolated data compared

## Table 1 | Tree species hyperdominance results for African, Amazonian and Southeast Asian tropical forests, resampled to the common sample size of 77,587 trees

| | Number of hyperdominants | Total species | Hyperdominant percentage | Fisher's α |
|---|---|---|---|---|
| **Africa** | 77 [62, 92] | 1,132 [1,069, 1,194] | 6.79 [5.39, 8.20] | 191 [161, 220] |
| **Amazonia** | 174 [134, 215] | 2,565 [2,419, 2,711] | 6.80 [5.24, 8.36] | 525 [475, 575] |
| **Southeast Asia** | 172 [125, 219] | 2,585 [2,440, 2,730] | 6.65 [4.59, 8.71] | 526 [476, 577] |

Numbers in brackets are confidence intervals derived from the s.d. across iterations of subsamples taken with replacement at the sample size of the Asia dataset. Resampling done by plot; 77,587 is the size of the Southeast Asia dataset.

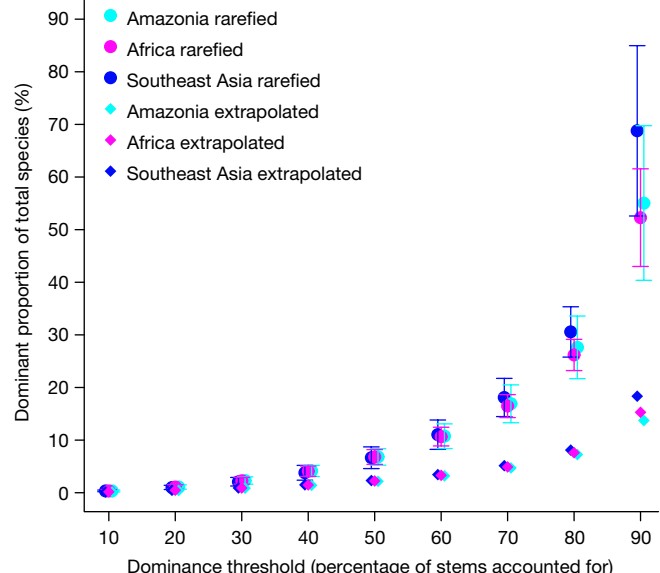

**Fig. 3 | The minimum percentage of total species required to account for given dominance thresholds of the total number of stems when this varies from 10% to 90%.** Circles show results as rarefied to the size of the Southeast Asia dataset (mean values across iterations of subsamples with 77,587 stems). Diamonds show the extrapolated results at the scale of the regions. Estimated rarefaction confidence intervals are derived from the s.d. across iterations of subsamples taken with replacement at 77,587 stems.

with those from the rarefied data are consistent with the pattern, described above, of many more rare species being added as the number of trees increases while many fewer common species are added (Fig. 2). Overall, the extrapolated results show that there are a tractable number of common species in tropical forests in Africa, Amazonia and Southeast Asia.

## Scaling to the tropics

We next estimate the number of common tropical tree species on Earth by multiplying the pan-tropical proportion of common species by the total number of tropical tree species on Earth. Our results suggest a pan-tropical hyperdominant percentage of 2.24% (Table 2). However, our extrapolations cannot provide an estimate of the total number of tropical tree species because we do not—for this study—have data from all tropical regions, including a lack of data from Central America, New Guinea and Micronesia. Furthermore, there is no consensus estimate of the total number of tropical tree species on Earth.

A compilation of lists of species known to science suggests a total of 60,065 tree species globally[15]. Tropical forest biomes likely comprise 63% of this list (E. Beech, personal communication, 2021), implying that there are around 37,900 known tropical tree species. This minimum estimate does not account for species that are yet to be identified and described by scientists. An alternative extrapolation method estimated that there are 46,900 species for the closed canopy tropical forest biome[25] (range 40,500–53,300 species), implying that there are 9,000 yet-to-be-identified species. This is in agreement with a recent global study suggesting that there are around 9,200 tree species remaining yet to be formally named, almost all in the tropics[16]. Thus, together, these studies suggest there are likely to be approximately 47,000 tropical tree species in the world's closed canopy tropical forests.

Our best estimate is that 1,053 tree species (2.24% of 47,000 species) account for half of Earth's 800 billion trees with trunk diameters of at least 10 cm found in the closed-canopy tropical forest biome. Although the true number may be lower or higher, the conclusion that a tractable

## Table 2 | Extrapolated tree species hyperdominance results for African, Amazonian, Southeast Asian tropical forests at the regional scale

| | Number of hyperdominants | Total species | Hyperdominant percentage |
|---|---|---|---|
| **Africa** | **104** [101, 107] | **4,638** [4,511, 4,764] | **2.23** |
| **Amazonia** | **299** [295, 304] | **13,826** [13,615, 14,036] | **2.16** |
| **Southeast Asia** | **278** [268, 289] | **11,963** [11,451, 12,475] | **2.32** |
| **Total**[a] | **681** [664, 700] | **30,427** [29,577, 31,275] | **2.24** |

[a]Calculated as the sum of the number of hyperdominants and total species across the three major tropical forest regions with hyperdominance percentage derived therefrom. Prediction intervals (in brackets) combine uncertainty from the standard error of predicted means and the residual s.d. of the regression of the bias correction fit.

number of species dominate tropical forests is clear. Some of these species are likely to be extraordinarily common: our best estimate is that just 61 species account for 80 billion individual trees (0.13% of 47,000 species). At the other end of the spectrum, we estimate that the rarest approximately 39,500 species account for just 80 billion trees, or 10% of individuals. Meanwhile, the other 90% of all trees are estimated to belong to just 7,487 species (15.93% of 47,000 species). Thus, these results open the possibility of focusing efforts on understanding the biology of a tractable number of species in tropical forests to approximate the whole stand.

## Identifying the most common species

Our analyses showing that 104, 299 and 278 common species account for 50% of the trees in our African, Amazonian and Southeast Asian study regions, respectively, do not yield a list of named species. To assess which named species are likely to be hyperdominant, we use a subsampling procedure similar to the rarefaction methodology above. We randomly subsample from approximately 10,000 trees per subsample (drawn by plot) and increase the size of the subsample in 10,000-tree increments until the size of each regional dataset is reached, and repeat this process 100 times. For each sampled increment of 10,000 trees we then calculate the proportion of random subsamples in which each species qualifies as hyperdominant (Supplementary Table 1). We then assign the species to one of four groups:

(1) Both hyperdominant in the full data and hyperdominant in the majority of subsamples even at very small sample sizes. These 50, 95 and 105 species in our Africa, Amazonia and Southeast Asia datasets, respectively, represent 3.5%, 2.1% and 4.1% of sampled species in each dataset. These species are likely to be geographically widespread and abundant.

(2) Both hyperdominant in the full data and hyperdominant in the majority of subsamples, but at the smallest sample sizes only occasionally hyperdominant. These 32, 129 and 67 species in our Africa, Amazonia and Southeast Asia datasets, respectively, represent 2.3%, 2.9% and 2.6% of sampled species in each dataset. These species are likely to be geographically widespread but not always abundant.

(3) Not quite hyperdominant in the full data, but hyperdominant in a substantial proportion of subsamples. These 102, 339 and 200 species in our Africa, Amazonia and Southeast Asia datasets, respectively, represent 7.2%, 7.5% and 7.7% of sampled species in each dataset. These species are probably locally abundant but not necessarily geographically widespread.

(4) Not hyperdominant in the full data and almost never hyperdominant in the subsamples. These 1,232, 3,929 and 2,213 species in our Africa, Amazonia and Southeast Asia datasets, respectively, represent 87%, 87.5% and 85.6% of sampled species in each dataset. These species are probably neither geographically widespread nor abundant.

We suggest that if all trees in a region were sampled, the hyperdominant species would be drawn from the first three groups, which are listed in Supplementary Table 2. This candidate list of 1,119 hyperdominant species contains 184 species in Africa, 563 species in Amazonia and 372 species in Southeast Asia, with no species appearing on more than one region's list. Thus, the list of species that are likely candidates for hyperdominance is manageably small.

There is uncertainty in our candidate hyperdominant list owing to the limitations of the underlying samples of plots across the landscape. Specifically, some species that always have low local abundance but are geographically widespread and lack habitat restrictions may require larger sample sizes for their hyperdominance to become clear. Similarly, species that combine low local abundance and habitat specificity pose challenges. If the distribution and extent of specialist habitat is great enough to result in hyperdominance of specialists but is not sufficiently captured in our sampling, such species might not appear in our candidate list. By contrast, some species in our candidate hyperdominant list will not be true hyperdominants. Of particular note, some apparently common species may actually comprise a group of cryptic species, with none of these cryptic species being hyperdominant by itself[31–33]. However, the striking similarly in species abundance patterns across the Africa, Amazonia and Southeast Asia datasets, despite differing sampling intensity on each continent, suggests that these potential limitations do not substantially affect the overall patterns found. We therefore expect a high overlap between our list of candidate hyperdominant species and eventual elucidation of the actual hyperdominants of these three regions and the pan-tropics.

Our list of 1,119 candidate hyperdominant species represents a tractable number of species on which to prioritize autecological research. Indeed, given their commonness, ecological data already exists for many of these species: 95% have some autecological data recorded in a large global database[34]; 83% have at least 10 different types of measurement, typically including their growth form, maximum height, wood density and aspects of leaf chemistry. This indicates that these species are already relatively well known. Therefore, only limited additional data may be required to open new approaches to better understanding tropical forests through their most common tree species, including how they may react to today's era of rapid global environmental change.

## Discussion

Charles Darwin wrote in *The Origin of Species* that "rarity is the attribute of a vast number of species of all classes and in all countries"[35]. If this is the case, then common species are themselves rare. Our results concur: despite their formidable diversity, the trees in tropical forests fit the 'rare is common, common is rare' pattern[36] which has been documented in many other taxa[17–19,36,37]. Beyond this, our analyses reveal highly consistent patterns of commonness across three major tropical forest regions. Notably, despite substantial inter-continental variation in biogeographic history, contemporary environment, forest structure and species composition, we have found an emergent property of the tropical forest system. For the trees that structure tropical forests, a consistent ~2.2% of the total species pool accounts for 50% of all individual trees in Africa, Amazonia and Southeast Asia. This consistency is all the more notable given relatively lower tree species richness of African tropical forests compared with Amazonian and Southeast Asian forests, probably owing to higher extinction rates in African forests, with evidence of major losses of African species at the Oligocene–Miocene boundary[38], and contractions of rainforest area due to drier conditions during repeated glacial–interglacial cycles over the past 2.6 million years[39].

We find common diversity patterns despite the very different histories of human occupancy in Amazonian, African and Southeast Asian tropical forests[40]. The relatively recent arrival of humans in Amazonia approximately 20,000 years ago has been linked to greater Pleistocene extinctions, in contrast to much longer human occupancy in the tropical forests of Africa and Southeast Asia[41]. Some have also suggested that Amazonian forest composition was altered by humans through the incipient domestication of tree species, increasing the abundance of a small number of favoured species[42]. Others have reported large areas of deforestation associated with the African Iron Age[43]. How can such different human histories result in near-identical patterns of tree species dominance? The most parsimonious explanation is that the system tends to return to a state with a similar species abundance pattern.

Nevertheless, consistent patterns of commonness do not necessarily imply the same causal mechanisms. The ubiquity of the broad 'rare is common, common is rare' pattern in ecology, which is also found in non-biological complex systems[44], means inferences as to the cause of this broad pattern are challenging[27,45]. Although combinatoric methods[45] and models that maximize the entropy of information[46,47] both produce the ubiquitous 'reverse lazy-J' pattern, empirical observations show fewer common species and more rare species than expected by statistical controls alone[45]. Similarly, neutral models produce the same broad pattern, but produce too few individuals of the most common Amazonian tree species[48]. This suggests that biological mechanisms influence tree community assembly to produce a consistent proportion of common species across continents.

Recent analyses have revealed that the same few families contribute most of the species richness in Africa and Amazonia[49], which when combined with analyses showing that more diverse families have more common species[50], may indicate a role for deep evolutionary mechanisms driving the patterns we find. Yet, considering the substantially smaller regional species pool in Africa compared with Amazonia and Southeast Asia, one might expect differing continental patterns of species dominance if evolutionary drivers were the primary mechanism, not the highly consistent patterns that we find. Similarly, if environmental filtering were a key mechanism, the different contemporary environments, with Africa much drier on average than the other two continents[26], and Southeast Asia consisting of scattered island-like areas of forest compared with the contiguous forested region of Amazonia, would also imply differing continental patterns of species dominance, not the near-identical patterns that we find. These constraints limit the potential mechanisms that could apply across our three-continent context.

One potential cross-continental mechanism is dispersal limitation, where the dispersal capabilities of species result in some suitable habitat patches remaining unoccupied. Another mechanism is density- or distance-dependent mortality, which appears widespread across tropical forests[51]. Here, specialist species-specific natural enemies such as pathogens and herbivores reduce seed or juvenile conspecific survival rates near conspecific adults or in areas of high juvenile conspecific density, thereby reducing competitive exclusion and contributing to the maintenance of high tree species richness in tropical forests[51]. It is possible that common species have largely evaded density- and/or distance-dependent mortality. Analyses showing that species abundance can be either high or low within given genera[52] support this hypothesis. Further progress on putative mechanisms can be made, for example, by exploring whether ecological or functional traits differ between common and rare species, and assessing the consistency of any differences among tropical continents[53]. Although deducing mechanisms is complex, the identification of a tractable number of common species in tropical forests will facilitate progress in understanding of tropical forests beyond species abundance distributions.

Refining our results, particularly the naming of common species, requires improved sampling of tropical forests, both in terms of geographic scope and taxonomic identification of trees within plots. Expanding sampling to include Central America, New Guinea, Micronesia and other regions would improve the generality of our results. Better identifying trees in existing plots would increase the utility of available samples: in our Southeast Asia region we excluded 142 plots

(approximately 120,000 stems) because they did not have more than 80% of trees identified to species. Furthermore, additional taxonomic research on even the most common species is needed given that some of the most common Amazonian[33] and African[54,55] tree species have been found to be complexes of several distinct species that are difficult to distinguish in the field. However, the similarity of our results across the three continental regions suggests that the occurrence of such species complexes may also be similar across the continental regions, again implying the operation of fundamental processes in differing forests. Overall, our work underscores the need for investment in taxonomy, particularly given the thousands of rare species we and others[18] document, but also when considering the most common species.

Our best estimate, using extrapolation, that for the tropics as a whole just 1,053 species account for half of Earth's 800 billion tropical trees has potentially profound implications. Rather than attempting to understand tens of thousands of species of tropical trees, a focus on just a few hundred of the most common species can provide a simplified characterization of these otherwise complex forests. Our analyses indicate that the most common of these species are reliably named and relatively well known. Our list of candidate hyperdominants can therefore readily serve new research, including in facilitating targeted autecological data collection to understand their role in providing ecological functions and services. Practically, this species-specific information could enhance tropical forest modelling by focusing on common species instead of relying on functional types or traits, thereby potentially improving predictions of future forest change.

In the future, analyses should be extended to investigate forest carbon stocks and hyperdominant species and their role in the provision of ecosystem services. In Amazonia, even fewer tree species were found to account for 50% of aboveground carbon stocks than the minimum number required to account for 50% of trees[22]. More generally, the set of common species is likely to include foundation species that define broader community assemblages, the environmental sensitivity of which will probably drive tropical forest responses to environmental change[56]. Of course, striving to understand and protect rare and non-hyperdominant species remains crucial, particularly as they face greater extinction risk and probably also contribute to the functioning of ecosystems, particularly when more functions[57], longer timescales[58] and imposed environmental changes[59] are considered, and given that the hyperdominants of the future may be rarer today. Nonetheless, with a complementary grasp of the most common species, mapping, understanding and modelling of the world's tropical forests will be a much more tractable proposition.

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

Declan L. M. Cooper[1,2]✉, Simon L. Lewis[1,3]✉, Martin J. P. Sullivan[3,4], Paulo I. Prado[5], Hans ter Steege[6,7], Nicolas Barbier[8,9], Ferry Slik[10], Bonaventure Sonké[9,11], Corneille E. N. Ewango[12], Stephen Adu-Bredu[13], Kofi Affum-Baffoe[14], Daniel P. P. de Aguiar[15,16], Manuel Augusto Ahuite Reategui[17], Shin-Ichiro Aiba[18], Bianca Weiss Albuquerque[19], Francisca Dionízia de Almeida Matos[20], Alfonso Alonso[21], Christian A. Amani[22,23], Dário Dantas do Amaral[24], Iêda Leão do Amaral[20], Ana Andrade[25], Ires Paula de Andrade Miranda[20], Ilondea B. Angoboy[26], Alejandro Araujo-Murakami[27], Nicolás Castaño Arboleda[28], Luzmila Arroyo[27], Peter Ashton[29], Gerardo A. Aymard C[30], Cláudia Baider[31,32], Timothy R. Baker[3], Michael Philippe Bessike Balinga[33], Henrik Balslev[34], Lindsay F. Banin[35], Olaf S. Bánki[6], Chris Baraloto[36], Edelcilio Marques Barbosa[20], Flávia Rodrigues Barbosa[37], Jos Barlow[38], Jean-Francois Bastin[39], Hans Beeckman[40], Serge Begne[3,9,11], Natacha Nssi Bengone[41], Erika Berenguer[38,42], Nicholas Berry[43], Robert Bitariho[44], Pascal Boeckx[45], Jan Bogaert[46], Bernard Bonyoma[47], Patrick Boundja[22,48], Nils Bourland[40,49,50,51], Faustin Boyemba Bosela[52], Fabian Brambach[53], Roel Brienen[3], David F. R. P. Burslem[54], José Luís Camargo[25], Wegliane Campelo[55], Angela Cano[56,57], Sasha Cárdenas[56], Dairon Cárdenas López[28], Rainiellen de Sá Carpanedo[37], Yrma Andreina Carrero Márquez[58], Fernanda Antunes Carvalho[59,60], Luisa Fernanda Casas[56], Hernán Castellanos[61], Carolina V. Castilho[62], Carlos Cerón[63], Colin A. Chapman[64,65,66], Jerome Chave[67], Phourin Chhang[68], Wanlop Chutipong[69], George B. Chuyong[70], Bruno Barçante Ladvocat Cintra[71], Connie J. Clark[72], Fernanda Coelho de Souza[59,73,74], James A. Comiskey[75,76], David A. Coomes[77], Fernando Cornejo Valverde[78], Diego F. Correa[56,79], Flávia R. C. Costa[59], Janaina Barbosa Pedrosa Costa[80], Pierre Couteron[8,9], Heike Culmsee[81], Aida Cuni-Sanchez[82,83], Francisco Dallmeier[21], Gabriel Damasco[84], Gilles Dauby[8,9], Nállarett Dávila[85], Hilda Paulette Dávila Doza[86], Jose Don T. De Alban[87,88], Rafael L. de Assis[89], Charles De Canniere[90], Thales De Haulleville[40], Marcelo de Jesus Veiga Carim[91], Layon O. Demarchi[19], Kyle G. Dexter[92,93], Anthony Di Fiore[94,95], Hazimah Haji Mohammad Din[96], Mathias I. Disney[1], Brice Yannick Djiofack[40,97,98], Marie-Noël K. Djuikouo[9,70], Tran Van Do[99], Jean-Louis Doucet[100], Freddie C. Draper[101], Vincent Droissart[8,9], Joost F. Duivenvoorden[102], Julien Engel[8,103], Vittoria Estienne[48], William Farfan-Rios[104,105], Sophie Fauset[106], Kenneth J. Feeley[107,108], Yuri Oliveira Feitosa[109], Ted R. Feldpausch[73,110], Cid Ferreira[20,293], Joice Ferreira[111], Leandro Valle Ferreira[24], Christine D. Fletcher[112], Bernardo Monteiro Flores[113], Alusine Fofanah[114], Ernest G. Foli[13], Émile Fonty[115,116], Gabriella M. Fredriksson[117], Alfredo Fuentes[105,118], David Galbraith[3], George Pepe Gallardo Gonzales[86], Karina Garcia-Cabrera[105], Roosevelt García-Villacorta[120,121], Vitor H. F. Gomes[122,123], Ricardo Zárate Gómez[124], Therany Gonzales[125], Rogerio Gribel[20], Marcelino Carneiro Guedes[80], Juan Ernesto Guevara[126,127], Khalid Rehman Hakeem[128], Jefferson S. Hall[129], Keith C. Hamer[130], Alan C. Hamilton[131], David J. Harris[93], Rhett D. Harrison[132], Terese B. Hart[133,134], Andy Hector[135], Terry W. Henkel[136], John Herbohn[137], Mireille B. N. Hockemba[48], Bruce Hoffman[138], Milena Holmgren[139], Euridice N. Honorio Coronado[140,141], Isau Huamantupa-Chuquimaco[142], Wannes Hubau[3,40,143], Nobuo Imai[144], Mariana Victória Irume[20], Patrick A. Jansen[145,146], Kathryn J. Jeffery[147], Eliana M. Jimenez[148], Tommaso Jucker[149], André Braga Junqueira[150], Michelle Kalamandeen[151], Narcisse G. Kamdem[9,11], Kuswata Kartawinata[152], Emmanuel Kasongo Yakusu[40,98,153], John M. Katembo[52], Elizabeth Kearsley[154], David Kenfack[129], Michael Kessler[155], Thiri Toe Khaing[156,157], Timothy J. Killeen[158], Kanehiro Kitayama[159], Bente Klitgaard[160], Nicolas Labrière[161], Yves Laumonier[161], Susan G. W. Laurance[162], William F. Laurance[162], Félix Laurent[40,97,98], Tinh Cong Le[163], Trai Trong Le[163], Miguel E. Leal[164], Evlyn Márcia Leão de Moraes Novo[165], Aurora Levesley[3], Moses B. Libalah[9,11,166], Juan Carlos Licona[167], Diógenes de Andrade Lima Filho[20], Jeremy A. Lindsell[168,169], Aline Lopes[170], Maria Aparecida Lopes[171], Jon C. Lovett[3,172], Richard Lowe[173], José Rafael Lozada[174], Xinghui Lu[175], Nestor K. Luambua[40,97,176,177], Bruno Garcia Luize[85], Paul Maas[6], José Leonardo Lima Magalhães[178,179], William E. Magnusson[59], Ni Putu Diana Mahayani[180], Jean-Remy Makana[181], Yadvinder Malhi[42], Lorena Maniguaje Rincón[20], Asyraf Mansor[182,183], Angelo Gilberto Manzatto[184], Beatriz S. Marimon[185], Ben Hur Marimon-Junior[185], Andrew R Marshall[82,186,187], Maria Pires Martins[20], Faustin M. Mbayu[188], Marcelo Brilhante de Medeiros[189], Italo Mesones[190], Faizah Metali[191], Vianet Mihindou[192,193], Jerome Millet[194], William Milliken[195], Hugo F. Mogollón[196], Jean-François Molino[8,9], Mohd. Nizam Mohd. Said[197], Abel Monteagudo Mendoza[188,198], Juan Carlos Montero[16,167], Sam Moore[42], Bonifacio Mostacedo[199], Linder Felipe Mozombite Pinto[86], Sharif Ahmed Mukul[137,200], Pantaleo K. T. Munishi[201], Hidetoshi Nagamasu[202], Henrique Eduardo Mendonça Nascimento[20], Marcelo Trindade Nascimento[203], David Neill[204], Reuben Nilus[205], Janaína Costa Noronha[37], Laurent Nsenga[40], Percy Núñez Vargas[198], Lucas Ojo[206], Alexandre A. Oliveira[5], Edmar Almeida de Oliveira[185], Fidèle Evouna Ondo[192], Walter Palacios Cuenca[207], Susamar Pansini[208], Marcelo Petratti Pansonato[20,32], Marcos Ríos Paredes[86], Ekananda Paudel[209], Daniela Pauletto[210], Richard G. Pearson[2], José Luis Marcelo Pena[211], R. Toby Pennington[93,110], Carlos A. Peres[212], Andrea Permana[213], Pascal Petronelli[214], Maria Cristina Peñuela Mora[215], Juan Fernando Phillips[216], Oliver L. Phillips[3], Georgia Pickavance[3], Maria Teresa Fernandez Piedade[19], Nigel C. A. Pitman[217], Pierre Ploton[8,9], Andreas Popelier[40,98,153], John R. Poulsen[72,218], Adriana Prieto[219], Richard B. Primack[220], Hari Priyadi[221], Lan Qie[3,222], Adriano Costa Quaresma[19,223], Helder Lima de Queiroz[224], Hirma Ramirez-Angulo[225], José Ferreira Ramos[20], Neidiane Farias Costa Reis[208], Jan Reitsma[226], Juan David Cardenas Revilla[20], Terhi Riutta[42,227], Gonzalo Rivas-Torres[95,228], Iyan Robiansyah[229,230], Maira Rocha[19], Domingos de Jesus Rodrigues[37], M. Elizabeth Rodriguez-Ronderos[87,231], Francesco Rovero[232,233], Andes H. Rozak[234], Agustín Rudas[235], Ervan Rutishauser[235], Daniel Sabatier[8,9], Le Bienfaiteur Sagang[9,11,236], Adeilza Felipe Sampaio[208], Ismayadi Samsoedin[237], Manichanh Satdichanh[209], Juliana Schietti[20], Jochen Schöngart[19], Veridiana Vizoni Scudeller[238], Naret Seuaturien[239], Douglas Sheil[240], Rodrigo Sierra[241], Miles R. Silman[242], Thiago Sanna Freire Silva[243], José Renan da Silva Guimarães[243], Murielle Simo-Droissart[9,11], Marcelo Fragomeni Simon[189], Plinio Sist[244], Thaiane R. Sousa[245], Emanuelle de Sousa Farias[246,247], Luiz de Souza Coelho[20], Dominick V. Spracklen[248], Suzanne M. Stas[248], Robert Steinmetz[239], Pablo R. Stevenson[56], Juliana Stropp[249], Rahayu S. Sukri[96], Terry C. H. Sunderland[137], Eizi Suzuki[18], Michael D. Swaine[252], Jianwei Tang[253], James Taplin[254], David M. Taylor[231], J. Sebastián Tello[255], John Terborgh[256,257], Nicolas Texier[258], Ida Theilade[259], Duncan W. Thomas[260], Raquel Thomas[261], Sean C. Thomas[262], Milton Tirado[241], Benjamin Toirambe[40,263], José Julio de Toledo[55], Kyle W. Tomlinson[156,264], Armando Torres-Lezama[225], Hieu Dang Tran[163], John Tshibamba Mukendi[40,153,265], Roven D. Tumaneng[88,266], Maria Natalia Umaña[267], Peter M. Umunay[268,269], Ligia Estela Urrego Giraldo[270], Elvis H. Valderrama Sandoval[271,272], Luis Valenzuela Gamarra[188], Tinde R. Van Andel[6,273], Martin van de Bult[274], Jaqueline van de Pol[275], Geertje van der Heijden[276], Rodolfo Vasquez[188], César I. A. Vela[277], Eduardo Martins Venticinque[278], Hans Verbeeck[279], Rizza Karen A. Veridiano[88,280], Alberto Vicentini[59], Ima Célia Guimarães Vieira[24], Emilio Vilanova Torre[225,268], Daniel Villarroel[27,281], Boris Eduardo Villa Zegarra[282], Jason Vleminckx[36,283], Patricio von Hildebrand[284], Vincent Antoine Vos[285], Corine Vriesendorp[217], Edward L. Webb[286,287], Lee J. T. White[41,147,288], Serge Wich[289], Florian Wittmann[16,223], Roderick Zagt[290], Runguo Zang[291], Charles Eugene Zartman[20], Lise Zemagho[9,11], Egleé L. Zent[292] & Stanford Zent[292]

[1]Department of Geography, University College London, London, UK. [2]Centre for Biodiversity and Environment Research, Department of Genetics, Evolution and Environment, University College London, London, UK. [3]School of Geography, University of Leeds, Leeds, UK. [4]Department of Natural Sciences, Manchester Metropolitan University, Manchester, UK. [5]Instituto de Biociências, Departamento de Ecologia, Universidade de Sao Paulo (USP), São Paulo, Brazil. [6]Naturalis Biodiversity Center, Leiden, The Netherlands. [7]Quantitative Biodiversity Dynamics, Department of Biology, Utrecht University, Utrecht, The Netherlands. [8]AMAP, Université de Montpellier, IRD, Cirad, CNRS, INRAE, Montpellier, France. [9]International Joint Laboratory DYCOFAC, IRD-UYI-IRGM, Yaoundé, Cameroon. [10]Environmental and Life Sciences, Faculty of Science, Universiti Brunei Darussalam, Gadong, Brunei Darussalam. [11]Plant Systematics and Ecology Laboratory, Higher Teachers' Training College, University of Yaoundé I, Yaoundé, Cameroon. [12]Faculty of Renewable Natural Resources Management and Faculty of Sciences, University of Kisangani, Kisangani, Democratic Republic of the Congo. [13]Forestry Research Institute of Ghana (FORIG), Kumasi, Ghana. [14]Mensuration Unit, Forestry Commission of Ghana, Kumasi, Ghana. [15]Procuradoria-Geral de Justiça, Ministério Público do Estado do Amazonas, Manaus, Brazil. [16]Instituto Nacional de Pesquisas da Amazônia (INPA), Manaus, Brazil. [17]Medio Ambiente, PLUSPRETOL, Iquitos, Peru. [18]Faculty of Environmental Earth Science, Hokkaido University, Sapporo, Japan. [19]Ecology, Monitoring and Sustainable Use of Wetlands (MAUA), Instituto Nacional de Pesquisas da Amazônia (INPA), Manaus, Brazil. [20]Coordenação de Biodiversidade, Instituto Nacional de Pesquisas da Amazônia (INPA), Manaus, Brazil. [21]Center for Conservation and Sustainability, Smithsonian Conservation Biology Institute, Washington, DC, USA. [22]Center for International Forestry Research (CIFOR), Bogor, Indonesia. [23]Université Officielle de Bukavu, Bukavu, Democratic Republic of the Congo. [24]Coordenação de Botânica, Museu Paraense Emílio Goeldi, Belém, Brazil. [25]Projeto Dinâmica Biológica de Fragmentos Florestais, Instituto Nacional de Pesquisas da Amazônia (INPA), Manaus, Brazil. [26]Institut National pour l'Etude et la Recherche Agronomiques, Bukavu, Democratic Republic of the Congo. [27]Museo de Historia Natural Noel Kempff Mercado, Universidad Autónoma Gabriel Rene Moreno, Santa Cruz, Santa Cruz, Bolivia. [28]Herbario Amazónico Colombiano, Instituto SINCHI, Bogotá, Colombia. [29]Bullard Emeritus Professor of Forestry, Harvard University, Cambridge, MA, USA. [30]Programa de Ciencias del Agro y el Mar, Herbario Universitario (PORT), UNELLEZ-Guanare, Guanare, Venezuela. [31]The Mauritius Herbarium, Agricultural Services, Ministry of Agro-Industry and Food Security, Reduit, Mauritius. [32]Instituto de Biociências, Departamento de Ecologia, Universidade de São Paulo (USP), São Paulo, Brazil. [33]Tetra Tech ARD, Accra, Ghana. [34]Department of Biology, Aarhus University, Aarhus C, Aarhus, Denmark. [35]UK Centre for Ecology and Hydrology, Penicuik, UK. [36]International Center for Tropical Botany, Department of Biological Sciences, Florida International University, Miami, FL, USA. [37]ICNHS, Federal University of Mato Grosso, Sinop, Brazil. [38]Lancaster Environment Centre, Lancaster University, Lancaster, UK. [39]TERRA Teaching and Research Centre, Gembloux Agro-Bio Tech, University of Liege, Gembloux, Belgium. [40]Service of Wood Biology, Royal Museum for Central Africa, Tervuren, Belgium. [41]Ministry of Forests, Seas, Environment and Climate, Libreville, Gabon. [42]Environmental Change Institute, School of Geography and the Environment, University of Oxford, Oxford, UK. [43]The Landscapes and Livelihoods Group, Edinburgh, UK. [44]Institute of Tropical Forest Conservation, Mbarara University of Science and Technology (MUST), Mbarara, Uganda. [45]Isotope Bioscience Laboratory (ISOFYS), Ghent University, Ghent, Belgium. [46]Biodiversity and Landscape Unit, Gembloux Agro-Bio Tech, Université de Liège, Liège, Belgium. [47]Section de la Foresterie, Institut National pour l'Etude et la Recherche Agronomique Yangambi, Yangambi, Democratic Republic of the Congo. [48]Congo Programme, Wildlife Conservation Society, Brazzaville, Republic of Congo. [49]CIFOR, Bogor, Indonesia. [50]Forest Resources Management, Gembloux Agro-Bio Tech, University of Liège, Liège, Belgium. [51]Resources and Synergies Development, Singapore, Singapore. [52]Laboratory of Ecology and Forest Management, Faculty of Sciences, University of Kisangani, Kisangani, Democratic Republic of the Congo. [53]Biodiversity, Macroecology and Biogeography, University of Göttingen, Göttingen, Germany. [54]School of Biological Sciences, University of Aberdeen, Aberdeen, UK. [55]Universidade Federal do Amapá, Ciências Ambientais, Macapá, Brazil. [56]Laboratorio de Ecología de Bosques Tropicales y Primatología, Universidad de los Andes, Bogotá, Colombia. [57]Cambridge University Botanic Garden, Cambridge, UK. [58]Programa de Maestria de Manejo de Bosques, Universidad de los Andes, Mérida, Mérida, Venezuela. [59]Coordenação de Pesquisas em Ecologia, Instituto Nacional de Pesquisas da Amazônia (INPA), Manaus, Brazil. [60]Departamento de Genética, Ecologia e Evolução, Instituto de Ciências Biológicas, Belo Horizonte, Brazil. [61]Centro de Investigaciones Ecológicas de Guayana, Universidad Nacional Experimental de Guayana, Puerto Ordaz, Venezuela. [62]Centro de Pesquisa Agroflorestal de Roraima, Embrapa Roraima, Boa Vista, Brazil. [63]Escuela de Biología Herbario Alfredo Paredes, Universidad Central, Quito, Ecuador. [64]Biology Department, Vancouver Island University, Nanaimo, British Columbia, Canada. [65]Shaanxi Key Laboratory for Animal Conservation,

Northwest University, Xi'an, China. [66]School of Life Sciences, University of KwaZulu-Natal, Scottsville, South Africa. [67]Laboratoire Évolution et Diversité Biologique, CNRS and Université Paul Sabatier, Toulouse, France. [68]Institute of Forest and Wildlife Research and Development (IRD), Phnom Penh, Cambodia. [69]Conservation Ecology Program, King Mongkut's University of Technology Thonburi, Bangkok, Thailand. [70]Faculty of Science, Department of Plant Science, University of Buea, Buea, Cameroon. [71]Instituto de Biociências, Departamento Botanica, Universidade de Sao Paulo (USP), São Paulo, Brazil. [72]Nicholas School of the Environment, Duke University, Durham, NC, USA. [73]University of Leeds, Leeds, UK. [74]BeZero, London, UK. [75]Inventory and Monitoring Program, National Park Service, Fredericksburg, VA, USA. [76]Smithsonian Conservation Biology Institute, Washington, DC, USA. [77]Department of Plant Sciences and Conservation Research Institute, University of Cambridge, Cambridge, UK. [78]Andes to Amazon Biodiversity Program, Madre de Dios, Madre de Dios, Peru. [79]The University of Queensland, Brisbane, Queensland, Australia. [80]Empresa Brasileira de Pesquisa Agropecuária, Embrapa Amapá, Macapá, Brazil. [81]State Agency for Environment, Nature Conservation and Geology, Güstrow, Germany. [82]Department of Environment and Geography, University of York, York, UK. [83]Department of International Environmental and Development Studies (NORAGRIC), Norwegian University of Life Sciences, Ås, Norway. [84]Gothenburg Global Biodiversity Centre, University of Gothenburg, Gothenburg, Sweden. [85]Departamento de Biologia Vegetal, Instituto de Biologia, Universidade Estadual de Campinas (UNICAMP), Campinas, Brazil. [86]Servicios de Biodiversidad EIRL, Iquitos, Peru. [87]Centre for Nature-Based Climate Solutions, Department of Biological Sciences, National University of Singapore, Singapore, Singapore. [88]Phillipines Programme, Fauna and Flora International, Cambridge, UK. [89]Biodiversity and Ecosystem Services, Instituto Tecnológico Vale, Belém, Brazil. [90]Landscape Ecology and Vegetal Production Systems Unit, Universite Libre de Bruxelles, Brussels, Belgium. [91]Departamento de Botânica, Instituto de Pesquisas Científicas e Tecnológicas do Amapá (IEPA), Macapá, Brazil. [92]School of Geosciences, University of Edinburgh, Edinburgh, UK. [93]Royal Botanic Garden Edinburgh, Edinburgh, UK. [94]Department of Anthropology, University of Texas at Austin, Austin, TX, USA. [95]Estación de Biodiversidad Tiputini, Colegio de Ciencias Biológicas y Ambientales, Universidad San Francisco de Quito (USFQ), Quito, Ecuador. [96]Institute for Biodiversity and Environmental Research, Universiti Brunei Darussalam, Bandar Seri Begawan, Brunei Darussalam. [97]Institut National pour l'Etude et la Recherche Agronomiques (INERA), Wood Laboratory of Yangambi, Yangambi, Democratic Republic of the Congo. [98]UGent-Woodlab, Laboratory of Wood Technology, Department of Environment, Faculty of Bioscience Engineering, Ghent University, Ghent, Belgium. [99]Silviculture Research Institute, Vietnamese Academy of Forest Sciences, Hanoi, Vietnam. [100]Forest Is Life, TERRA, Gembloux Agro-Bio Tech, Liège University, Liège, Belgium. [101]Department of Geography and Planning, University of Liverpool, Liverpool, UK. [102]Institute of Biodiversity and Ecosystem Dynamics, University of Amsterdam, Amsterdam, The Netherlands. [103]Florida International University, Miami, FL, USA. [104]Living Earth Collaborative, Washington University in Saint Louis, St Louis, MO, USA. [105]Missouri Botanical Garden, St Louis, MO, USA. [106]School of Geography, Earth and Environmental Sciences, University of Plymouth, Plymouth, UK. [107]Department of Biology, University of Miami, Coral Gables, FL, USA. [108]Fairchild Tropical Botanic Garden, Coral Gables, FL, USA. [109]Programa de Pós-Graduação em Biologia (Botânica), Instituto Nacional de Pesquisas da Amazônia (INPA), Manaus, Brazil. [110]Department of Geography, College of Life and Environmental Sciences, University of Exeter, Exeter, UK. [111]Empresa Brasileira de Pesquisa Agropecuária, Embrapa Amazônia Oriental, Belém, Brazil. [112]Forest Research Institute Malaysia, Kepong, Malaysia. [113]Postgraduate Program in Ecology, Federal University of Santa Catarina, Florianópolis, Brazil. [114]The Gola Rainforest National Park, Kenema, Sierra Leone. [115]Direction Régionale de la Guyane, Office National des Forêts, Cayenne, French Guiana. [116]Université de Montpellier, Montpellier, France. [117]Pro Natura Foundation, Balikpapan, Indonesia. [118]Herbario Nacional de Bolivia, Instituto de Ecología, Carrera de Biología, Universidad Mayor de San Andrés, La Paz, Bolivia. [119]Biology Department and Center for Energy, Environment and Sustainability, Wake Forest University, Winston Salem, NC, USA. [120]Programa Restauración de Ecosistemas (PRE), Centro de Innovación Científica Amazónica (CINCIA), Tambopata, Peru. [121]Peruvian Center for Biodiversity and Conservation (PCBC), Iquitos, Peru. [122]Escola de Negócios Tecnologia e Inovação, Centro Universitário do Pará, Belém, Brazil. [123]Universidade Federal do Pará, Belém, Brazil. [124]PROTERRA, Instituto de Investigaciones de la Amazonía Peruana (IIAP), Iquitos, Peru. [125]ACEER Foundation, Puerto Maldonado, Peru. [126]Grupo de Investigación en Biodiversidad, Medio Ambiente y Salud-BIOMAS, Universidad de las Américas, Quito, Ecuador. [127]The Field Museum, Chicago, IL, USA. [128]Department of Biological Sciences, Faculty of Science, King Abdulaziz University, Jeddah, Saudi Arabia. [129]Forest Global Earth Observatory (ForestGEO), Smithsonian Tropical Research Institute, Washington, DC, USA. [130]School of Biology, University of Leeds, Leeds, UK. [131]Honorary Professor, Kunming Institute of Botany, Chinese Academy of Science, Kunming, China. [132]World Agroforestry, Lusaka, Zambia. [133]Lukuru Wildlife Research Foundation, Kinshasa, Democratic Republic of the Congo. [134]Division of Vertebrate Zoology, Yale Peabody Museum of Natural History, New Haven, CT, USA. [135]Department of Plant Sciences, University of Oxford, Oxford, UK. [136]Department of Biological Sciences, California State Polytechnic University, Humboldt, Arcata, CA, USA. [137]Tropical Forests and People Research Centre, University of the Sunshine Coast, Maroochydore DC, Queensland, Australia. [138]Amazon Conservation Team, Arlington, USA. [139]Resource Ecology Group, Wageningen University and Research, Wageningen, The Netherlands. [140]Instituto de Investigaciones de la Amazonía Peruana (IIAP), Iquitos, Peru. [141]University of St Andrews, St Andrews, UK. [142]Herbario HAG, Universidad Nacional Amazónica de Madre de Dios (UNAMAD), Puerto Maldonado, Peru. [143]Department of Environment, Laboratory of Wood Technology (Woodlab), Ghent University, Ghent, Belgium. [144]Department of Forest Science, Tokyo University of Agriculture, Tokyo, Japan. [145]Smithsonian Tropical Research Institute, Ancon, Panama. [146]Department of

Environmental Sciences, Wageningen University and Research, Wageningen, The Netherlands. [147]Department of Biological and Environmental Sciences, University of Stirling, Stirling, UK. [148]Grupo de Ecología y Conservación de Fauna y Flora Silvestre, Instituto Amazónico de Investigaciones Imani, Universidad Nacional de Colombia sede Amazonia, Leticia, Colombia. [149]School of Biological Sciences, University of Bristol, Bristol, UK. [150]Institut de Ciència i Tecnologia Ambientals, Universitat Autònoma de Barcelona, Barcelona, Spain. [151]School of Earth, Environment and Society, McMaster University, Hamilton, Ontario, Canada. [152]Integrative Research Center, The Field Museum of Natural History, Chicago, IL, USA. [153]Faculté de Gestion de Ressources Naturelles Renouvelables, Université de Kisangani, Kisangani, Democratic Republic of the Congo. [154]Computational and Applied Vegetation Ecology (CAVElab), Department of Environment, Faculty of Bioscience Engineering, Ghent University, Ghent, Belgium. [155]Department of Systematic and Evolutionary Botany, University of Zurich, Zurich, Switzerland. [156]Center for Integrative Conservation, Xishuangbanna Tropical Botanical Garden, Chinese Academy of Sciences, Menglun, Mengla, China. [157]University of the Chinese Academy of Sciences, Beijing, China. [158]Agteca—Amazonica, Santa Cruz, Bolivia. [159]Graduate School of Agriculture, Kyoto University, Kyoto, Japan. [160]Department for Accelerated Taxonomy, Royal Botanic Gardens, Richmond, UK. [161]Forest and Environment Program, Center for International Forestry Research (CIFOR), Bogor, Indonesia. [162]Centre for Tropical Environmental and Sustainability Science and College of Science and Engineering, James Cook University, Cairns, Queensland, Australia. [163]Viet Nature Conservation Centre, Hanoi, Viet Nam. [164]Uganda Programme, Wildlife Conservation Society, Kampala, Uganda. [165]Divisao de Sensoriamento Remoto (DSR), Instituto Nacional de Pesquisas Espaciais (INPE), São José dos Campos, Brazil. [166]Department of Plant Biology, Faculty of Science, University of Yaoundé I, Yaoundé, Cameroon. [167]Instituto Boliviano de Investigacion Forestal, Santa Cruz, Santa Cruz, Bolivia. [168]The RSPB, Sandy, UK. [169]A Rocha International, Cambridge, UK. [170]Department of Ecology, Institute of Biological Sciences, University of Brasilia, Brasilia, Brazil. [171]Instituto de Ciências Biológicas, Universidade Federal do Pará, Belém, Brazil. [172]Herbarium, Royal Botanic Gardens Kew, Richmond, UK. [173]Botany Department, University of Ibadan, Ibadan, Nigeria. [174]Facultad de Ciencias Forestales y Ambientales, Instituto de Investigaciones para el Desarrollo Forestal, Universidad de los Andes, Mérida, Mérida, Venezuela. [175]Institute of Forest Ecology, Environment and Protection, Chinese Academy of Forestry, Beijing, China. [176]Faculty of Renewable Natural Resources Management, University of Kisangani, Kisangani, Democratic Republic of the Congo. [177]Faculté des sciences Agronomiques, Université Officielle de Mbujimayi, Mbujimayi, Democratic Republic of the Congo. [178]Programa de Pós-Graduação em Ecologia, Universidade Federal do Pará, Belém, Brazil. [179]Embrapa Amazônia Oriental, Belém, Brazil. [180]Faculty of Forestry, Universitas Gadjah Mada, Yogyakarta, Indonesia. [181]Faculté des Sciences, Laboratoire d'Écologie et Aménagement Forestier, Université de Kisangani, Kisangani, Democratic Republic of the Congo. [182]School of Biological Sciences, Universiti Sains Malaysia, George Town, Malaysia. [183]Centre for Marine and Coastal Studies, Universiti Sains Malaysia, George Town, Malaysia. [184]Departamento de Biologia, Universidade Federal de Rondônia, Unir, Porto Velho, Brazil. [185]Programa de Pós-Graduação em Ecologia e Conservação, Universidade do Estado de Mato Grosso, Nova Xavantina, Brazil. [186]Flamingo Land, Kirby Misperton, UK. [187]Forest Research Institute, University of the Sunshine Coast, Sippy Downs, Queensland, Australia. [188]Jardín Botánico de Missouri, Oxapampa, Peru. [189]Embrapa Recursos Genéticos e Biotecnologia, Brasilia, Brazil. [190]Department of Integrative Biology, University of California, Berkeley, CA, USA. [191]Environmental and Life Sciences Programme, Faculty of Science, Universiti Brunei Darussalam, Bandar Seri Begawan, Brunei Darussalam. [192]Agence Nationale des Parcs Nationaux, Libreville, Gabon. [193]Ministère de la Forêt, de la Mer, de l'Environnement, Chargé du Plan Climat, Libreville, Gabon. [194]Office français de la biodiversité, Vincennes, France. [195]Department for Ecosystem Stewardship, Royal Botanic Gardens, Richmond, UK. [196]Endangered Species Coalition, Silver Spring, MD, USA. [197]Institute of Climate Change, Universiti Kebangsaan Malaysia, Bangi, Malaysia. [198]Herbario Vargas, Universidad Nacional de San Antonio Abad del Cusco, Cuzco, Peru. [199]Facultad de Ciencias Agrícolas, Universidad Autónoma Gabriel René Moreno, Santa Cruz, Santa Cruz, Bolivia. [200]Department of Environment and Development Studies, United International University, Dhaka, Bangladesh. [201]Department of Ecosystems and Conservation, Sokoine University of Agriculture, Morogoro, Tanzania. [202]The Kyoto University Museum, Kyoto University, Kyoto, Japan. [203]Laboratório de Ciências Ambientais, Universidade Estadual do Norte Fluminense, Campos dos Goytacazes, Brazil. [204]Universidad Estatal Amazónica, Puyo, Ecuador. [205]Forest Research Centre, Sandakan, Malaysia. [206]University of Abeokuta, Abeokuta, Nigeria. [207]Herbario Nacional del Ecuador, Universidad Técnica del Norte, Quito, Ecuador. [208]Programa de Pós-Graduação em Biodiversidade e Biotecnologia PPG-Bionorte, Universidade Federal de Rondônia, Porto Velho, Brazil. [209]Centre for Mountain Ecosystem Studies, Kunming Institute of Botany, Chinese Academy of Sciences, Kunming, China. [210]Instituto de Biodiversidade e Florestas, Universidade Federal do Oeste do Pará, Santarém, Brazil. [211]Universidad Nacional de Jaén, Cajamarca, Peru. [212]School of Environmental Sciences, University of East Anglia, Norwich, UK. [213]University of Warwick, Warwick, UK. [214]Cirad UMR Ecofog, AgrosParisTech, CNRS, INRAE, Université Guyane, Kourou Cedex, France. [215]Universidad Regional Amazónica IKIAM, Tena, Ecuador. [216]Fundación Puerto Rastrojo, Bogotá, Colombia. [217]Science and Education, The Field Museum, Chicago, IL, USA. [218]The Nature Conservancy, Boulder, CO, USA. [219]Instituto de Ciencias Naturales, Universidad Nacional de Colombia, Bogotá, Colombia. [220]Biology Department, Boston University, Boston, MA, USA. [221]Department of Resource and Environmental Economics (ESL), IPB University, Bogor, Indonesia. [222]School of Life Sciences, University of Lincoln, Lincoln, UK. [223]Wetland Department, Institute of Geography and Geoecology, Karlsruhe Institute of Technology (KIT), Rastatt, Germany. [224]Diretoria Técnico-Científica, Instituto de Desenvolvimento Sustentável Mamirauá, Tefé, Brazil. [225]Instituto de Investigaciones para el Desarrollo Forestal (INDEFOR), Universidad de los Andes, Mérida, Mérida, Venezuela.

[226]Waardenburg Ecology, Culemborg, The Netherlands. [227]College of Life Sciences, University of Exeter, Exeter, UK. [228]University of Florida, Gainesville, FL, USA. [229]Department of Biological Sciences, King Abdulaziz University, Jeddah, Kingdom of Saudi Arabia. [230]Center for Plant Conservation Bogor Botanic Gardens, Indonesian Institute of Science, Bogor, Indonesia. [231]Department of Geography, National University of Singapore, Singapore, Singapore. [232]Deparment of Biology, University of Florence, Sesto Fiorentino, Italy. [233]Tropical Biodiversity Section, Museo delle Scienze (MUSE), Trento, Italy. [234]Research Center for Plant Conservation, Botanic Gardens and Forestry, National Research and Innovation Agency (BRIN), Bogor, Indonesia. [235]InfoFlora, Botanical Garden of Geneva, Geneva, Switzerland. [236]Institute of the Environment and Sustainability, University of California, Los Angeles, CA, USA. [237]Forest Research and Development Center, Research, Development and Innovation Agency, Ministry of Environment and Forestry, Bogor, Indonesia. [238]Departamento de Biologia, Universidade Federal do Amazonas (UFAM)–Instituto de Ciências Biológicas (ICB1), Manaus, Brazil. [239]World Wildlife Fund Thailand, Bangkok, Thailand. [240]Forest Ecology and Forest Management Group, Wageningen University and Research, Wageningen, The Netherlands. [241]GeoIS, Quito, Ecuador. [242]Biological and Environmental Sciences, University of Stirling, Stirling, UK. [243]Amcel Amapá Florestal e Celulose SA, Santana, Brazil. [244]Cirad-ES, Campus International de Baillarguet, TA C-105/D, Montpellier, France. [245]Programa de Pós-Graduação em Ecologia, Instituto Nacional de Pesquisas da Amazônia (INPA), Manaus, Brazil. [246]Laboratório de Ecologia de Doenças Transmissíveis da Amazônia (EDTA), Instituto Leônidas e Maria Deane, Fiocruz, Manaus, Brazil. [247]Instituto Oswaldo Cruz (IOC/FIOCRUZ), Rio de Janeiro, Brazil. [248]School of Earth and Environment, University of Leeds, Leeds, UK. [249]Biogeography Department, Trier University, Trier, Germany. [250]Faculty of Forestry, University of British Columbia, Vancouver, British Columbia, Canada. [251]Research Center for the Pacific Islands, Kagoshima University, Kagoshima, Japan. [252]Department of Plant and Soil Science, School of Biological Sciences, University of Aberdeen, Aberdeen, UK. [253]Key Laboratory of Tropical Plant Resources and Sustainable Use, Xishuangbanna Tropical Botanical Garden, Chinese Academy of Sciences, Mengla, China. [254]UK Research and Innovation, Innovate UK, London, UK. [255]Center for Conservation and Sustainable Development, Missouri Botanical Garden, St Louis, MO, USA. [256]Department of Biology and Florida Museum of Natural History, University of Florida, Gainesville, FL, USA. [257]James Cook University, Cairns, Queensland, Australia. [258]Université Libre de Bruxelles, Brussels, Belgium. [259]Department of Food and Resource Economics, University of Copenhagen, Copenhagen, Denmark. [260]School of Biological Sciences, Washington State University, Vancouver, WA, USA. [261]Iwokrama International Centre for Rain Forest Conservation and Development, Georgetown, Guyana. [262]Institute of Forestry and Conservation, University of Toronto, Toronto, Ontario, Canada. [263]Ministère de l'Environnement et Développement Durable, Kinshasa, Democratic Republic of the Congo. [264]Center of Conservation Biology, Core Botanical Gardens, Chinese Academy of Sciences, Menglun, China. [265]Faculté des Sciences Appliquées, Université de Mbujimayi, Mbujimayi, Democratic Republic of the Congo. [266]Emerging Technology Development Division, Department of Science and Technology Philippine Council for Industry, Energy and Emerging Technology Research and Development (DOST-PCIEERD), Taguig City, Philippines. [267]Department of Ecology and Evolutionary Biology, University of Michigan, Ann Arbor, MI, USA. [268]Wildlife Conservation Society, New York, NY, USA. [269]Yale School of Forestry and Environmental Studies, Yale University, New Haven, CT, USA. [270]Departamento de Ciencias Forestales, Universidad Nacional de Colombia, Medellín, Colombia. [271]Department of Biology, University of Missouri, St Louis, MO, USA. [272]Universidad Nacional de la Amazonia Peruana, Iquitos, Peru. [273]Wageningen University, Wageningen, The Netherlands. [274]Doi Tung Development Project, Social Development Department, Chiang Rai, Thailand. [275]Compagnie des Bois du Gabon, Port Gentil, Gabon. [276]University of Nottingham, Nottingham, UK. [277]Escuela Profesional de Ingeniería Forestal, Universidad Nacional de San Antonio Abad del Cusco, Puerto Maldonado, Peru. [278]Centro de Biociências, Departamento de Ecologia, Universidade Federal do Rio Grande do Norte, Natal, Brazil. [279]CAVElab—Computational and Applied Vegetation Ecology, Department of Environment, Ghent University, Ghent, Belgium. [280]FORLIANCE, Bonn, Germany. [281]Fundación Amigos de la Naturaleza (FAN), Santa Cruz, Bolivia. [282]Direccíon de Evaluación Forestal y de Fauna Silvestre, Magdalena del Mar, Peru. [283]Faculté des Sciences, Service d'Évolution Biologique et Écologie, Université Libre de Bruxelles, Brussels, Belgium. [284]Fundación Estación de Biología, Bogotá, Colombia. [285]Instituto de Investigaciones Forestales de la Amazonía, Universidad Autónoma del Beni José Ballivián, Riberalta, Beni, Bolivia. [286]Viikki Tropical Resources Institute, Department of Forest Sciences, University of Helsinki, Helsinki, Finland. [287]Helsinki Institute of Sustainability Science (HELSUS), Helsinki, Finland. [288]Institut de Recherche en Écologie Tropicale, Libreville, Gabon. [289]School of Biological and Environmental Sciences, Liverpool John Moores University, Liverpool, UK. [290]Tropenbos International, Ede, The Netherlands. [291]Key Laboratory of Forest Ecology and Environment of State Forestry Administration, Institute of Forest Ecology, Environment and Protection, Chinese Academy of Forestry, Beijing, China. [292]Laboratory of Human Ecology, Instituto Venezolano de Investigaciones Científicas (IVIC), Caracas, Venezuela. [293]Deceased: Cid Ferreira.

## Methods

### Data compilation and pre-processing

We collated data from forest inventory plots ≥0.2 ha in size, situated in structurally intact (no detectable past logging or fire), closed canopy (not dry forest or savanna) tropical forest, with enumeration of all stems ≥10 cm diameter, in which ≥ 80% of stems are identified to the species level. Following Sullivan et al.[61], small (≤0.5 ha) plots within 1 km of each other were grouped for analysis to minimize the effect of stochastic tree fall events in smaller areas[62]. These criteria allow direct comparisons to be made with hyperdominance results from Amazonia[6,21]. The data from each continent comprise the following:

Africa: 483 plots, covering a total of 504 ha (mean plot area 1.04 ha, median 1 ha, range 0.2–10 ha). These data are from four sources: 299 plots from the African Tropical Rainforest Observatory Network[1,63] (AfriTRON: www.afritron.org, accessed 1 March 2020), curated at http://www.ForestPlots.net[64]; 127 plots from the Central African Plot Network (https://central-african-plot-network.netlify.app); 52 plots from the TEAM network[2]; and 5 × 1 ha plots from 5 different soil types, extracted from one 50-ha plot in Korup, Cameroon from the SIGEO/CTFS network[3].

Amazonia: 1,417 plots, covering a total of 1,591 ha (mean plot area 1.12 ha, median 1 ha, range 0.1–78.8 ha) from the Amazon Tree Diversity Network (ATDN: http://atdn.myspecies.info/, includes plots from the RAINFOR network), accessed 8 January 2020.

Southeast Asia: 230 plots, covering a total of 202 ha (mean plot area 0.88 ha, median 0.49 ha, range 0.21–4.5 ha). These data are from two sources: 143 plots from Slik et al.[4,25]—a decrease from the published Indo-Pacific dataset in Slik et al.[4,25] due to our ≥80% species identification criterion and our Southeast Asia study region excluding Australia, India, and Papua New Guinea; and 87 plots from the T-Forces network[64] curated at http://www.ForestPlots.net, accessed 03/02/2021.

Species names were checked for orthography and standardized (synonyms identified from the reference databases corrected to their accepted names) using the African Flowering Plants Database (https://www.ville-ge.ch/musinfo/bd/cjb/africa), Taxonomic Name Resolution Service[65], and Asian Plant Synonym Lookup (F. Slik, personal communication), for Africa, Amazonia and Southeast Asia, respectively. Trees not identified to species level (7.3%, 6.3% and 8.4% of stems in the Africa, Southeast Asia datasets respectively) were classed as 'indeterminate' (Indet). Indet stems contributed to plot-level and dataset-wide stem abundance totals but are necessarily absent from species totals.

For the purposes of our study we delimited tropical forests according to the 'tropical and subtropical moist broadleaf forests' biome delineation from the World Wildlife Fund ecoregion map[60]. The total number of tropical trees ≥10 cm trunk diameter in each of our regions was then estimated by summing tree abundances in countries in which we have at least one sampled plot from the 'map of Global Tree Density'[66] (derived from 429,775 ground-based estimates of tree density) and masking according to the 'tropical and subtropical moist broadleaf forests' borders using ArcGIS v3.10.1[67]. Thus, we estimate that there are ~92 billion, ~331 billion trees, and ~217 billion trees in our Africa, Amazonia, and Southeast Asia regions, respectively, totalling 640 billion trees. Including abundance from countries in the 'tropical and subtropical moist broadleaf forests' biome in which we have no sampled plots, we estimate ~799 billion total trees across all of Earth's moist tropical forests.

### Data format, commonness and diversity parameters

The species abundance distribution (SAD), defined as a vector of abundances (number of individuals observed) of all species encountered in a community[17], formed the basis for our analyses of the three tropical forest datasets. For each dataset, we tallied the number of trees of each species in each plot to give plot-level SADs and combined these SADs across all plots to get regional-level abundance matrices with rows representing plots, columns representing species, and entries representing the abundance of each species in each plot. To capture patterns of commonness and species composition we calculated the number of hyperdominants (H#), defined as the minimum number of species required to account for 50% of the population of an assemblage[6], hyperdominant species identities, total number of species (TS), hyperdominant percentage of total species (H% = H#/TS) and Fisher's α (ref. 68). To investigate the sensitivity of results to the 'hyperdominant' definition of the most common species, we looked beyond the 50% threshold used for hyperdominance, at the minimum number of species required to account for 10%, 20%, 30%, ..., 90% of the population, here termed 'dominants'.

### Sampling standardization, subsampling and comparison of continental data

We identified variations in the number of plots, stems, and species, and the size and spatial clustering of plots as potential confounding factors liable to skew dominance and diversity results from our regional datasets and impede rigorous comparisons between them. We used sample-based rarefaction to quantify and account for the effect of differences in sample size (number of plots and stems) on our diversity measures of interest; namely species richness, number, ranking and identity of hyperdominants, hyperdominant percentage of total species, and Fisher's α. To quantify the effect of plot size, which is smaller in Southeast Asia data (mean 0.88 ha, median 0.49 ha) than in Amazonia and Africa data (both mean ~1 ha, median 1 ha) we compared results from the full data to those from plots >0.9 ha. We found that small plots (≪1 ha) inflate per-plot species totals relative to larger plots (because the rate of encountering new species is higher the smaller the plot size; Extended Data Fig. 1), so we limited our analyses to plots >0.9 ha to enable like-for-like comparison.

For Africa, we retained 368 plots covering 450 ha (mean plot area 1.22 ha, median 1 ha, range 0.92–10 ha; 2% of plots 0.9–0.99 ha, 88% of plots 1 ha, 8% of plots 1.01–5 ha, 1% of plots >5 ha) with mean temperature of 24.3 °C (range 16.2–27.6 °C), mean annual precipitation 1,802 mm yr$^{-1}$, (range 1,066–2,747 mm yr$^{-1}$), and mean elevation of 511 m above sea level (range 41–2,070 m) per WorldClim[69]. For Amazonia we retained 1,097 plots covering 1,434 ha (mean plot area 1.31 ha, median 1 ha, range 0.9–78.8 ha; 2% of plots 0.9–0.99 ha, 90% of plots 1 ha, 7% of plots 1.01–5 ha, 1% of plots >5 ha) with mean temperature of 26.0 °C (range 20.9–27.6 °C), mean annual precipitation 2,397 mm yr$^{-1}$ (range 1,119–4,284 mm yr$^{-1}$), and mean elevation of 154 m (range 0–1,142 m). For Southeast Asia we retained 103 plots covering 164 ha (mean plot area 1.59 ha, median 1 ha, range 0.96–4.5 ha; 1% of plots 0.9–0.99 ha, 48% of plots 1 ha, 52% of plots 1.01–5 ha, 0% of plots >5 ha) with mean temperature of 25.7 °C (range 20.1–27.5 °C), mean precipitation 2,680 mm yr$^{-1}$ (range 1,466–3,941 mm yr$^{-1}$), and mean elevation of 288 m (range 10–934 m). We assessed if the remaining differences in plot size affected the results, using only the 1 ha plots from Africa ($n$ = 323) and Amazonia ($n$ = 988), rarefied to the size of the Asia dataset, again finding near-identical per cent hyperdominance on the two continents (Africa: 7.30%, 95% confidence interval: 6.56–8.04; Amazonia: 7.35%, 95% confidence interval: 6.61–8.10).

To quantify the effect of the spatial clustering of plots, we compared results from the full Amazonia data, as the largest dataset, to those from subsets of the Amazonia data in which 1,2,3,...,10 plots were sampled from each spatial cluster. We found that spatial clustering had a negligible and not statistically significant effect on hyperdominant percentage and fitted values of Fisher's α (Extended Data Fig. 2). Therefore, we retain all plots for our analyses to maximize sample sizes. Computation of percentage hyperdominance and dominance accounts for the effects of variations in species richness on the number of hyperdominants and dominants.

For sample-based rarefaction, 200 subsamples of 1, 2, ..., $N_p$ plots were drawn, without replacement, from the $N_p$ total number of plots in the $p$th dataset, the stems contained in each subsample were pooled,

and the mean total species, number of hyperdominants, hyperdominance percentage, and Fisher's α were calculated across the subsamples. Similarly, we tallied the number of subsamples in which each species in the dataset qualified as hyperdominant at each level of subsampling and compared results between datasets at subsample sizes equating to a mean 10,000, 20,000, …, $I_p$ individual trees, where $I_p$ is the total number of trees in the $p$th dataset. Confidence intervals were calculated as confidence interval = $\mu \pm 1.96 \times \sigma$, where $\mu$ values are the means of the diversity metrics calculated across the 200 iterations of subsamples taken without replacement, and $\sigma$ values are the s.d. of the mean of diversity metrics calculated across the 200 iterations of subsamples taken with replacement (to reduce the degree to which confidence intervals were conditional on the sample). For point estimates, all datasets were compared at the common sample size of the Southeast Asia dataset (77,587 stems equivalent to 150, 116 and 103 plots in Africa, Amazonia and Southeast Asia, respectively).

### Extrapolation and bias correction of log series fits to the empirical data

We extrapolated our empirical SADs to SADs at the scale of the entire Amazonian, African, and Southeast Asian regional level via analytical expansion and bias correction of Fisher's log series fits following the methodology of ter Steege et al.[21] developed using the ATDN data that comprise our Amazonia dataset.

Ter Steege[21] et al. found that simulations of sampling of plots with conspecific aggregation from log series-modelled SADs provide extremely good approximations of the processes that generate tropical forest inventory data—that is, non-random sampling of plots containing species with limited dispersal and/or ecological preferences. They further found that estimates of species richness derived from samples taken with conspecific aggregation from the simulated SADs substantially underestimated the true species richness of the simulated SADs, but that a linear relationship with low variance existed between the true and sample-derived values. Thus, although conspecific aggregation in the empirical data introduces bias in the log series-modelled SADs extrapolated therefrom, quantification and correction of the effects of this bias on regional estimates of species richness is possible. Therefore, to estimate species richness at the regional level, they fitted Fisher's log series to empirical species abundance data, quantified the effect of conspecific aggregation on these estimates via simulation, and applied quantified corrections to give more accurate estimates of regional species richness taking into conspecific aggregation. Thus, this approach corrects for species-specific aggregation at the plot scale depending on species density.

To estimate regional numbers and proportions of dominants and hyperdominants as well as species richness, we extended the methodology of ter Steege et al.[21] to log series-derived estimates of regional numbers and proportions of dominants and hyperdominants. Initially, values of Fisher's α were fitted to the empirical species abundance vectors from each region using maximum likelihood and numerical optimization in the 'sads' R package[70] and fits visualized with Preston plots[71] and rank abundance distributions (RAD)[36] (Extended Data Fig. 4). Regional species totals $S$, not accounting for bias introduced by conspecific aggregation, were then estimated[68] via $S = \alpha \times \ln\left(1 + \frac{N}{\alpha}\right)$ with total number of trees ≥10 cm trunk diameter at the continental level ($N$) from the Global Tree Density map of Crowther et al.[66] with each tropical region delineated within the 'tropical and subtropical moist broadleaf forests' biome of Olson et al.[60]. An inverse quantile function from the sads R package[70] was then applied to generate (uncorrected) continental-scale SADs for each region using the above fitted $\alpha$, estimated $S$ and $N$.

For the quantification of bias and computation of corrections, we first simulated 250 log series SADs with known values of total species, $S_k$, randomly drawn from the range of plausible regional species totals (10,000–25,000 in Amazonia and Southeast Asia; 2,000–10,000 in

Africa) and $N$, the number of trees in each region ≥10 cm trunk diameter from Crowther et al.[66]. We calculated known values of numbers of hyperdominants, $H_k$, and percentage hyperdominance, $P_k$, from each of these simulated distributions. Using a negative binomial distribution to simulate conspecific aggregation per ter Steege et al.[21], we then simulated $j$ random samples of 1-ha plots from each of the 250 simulated SADs, with $j$ equal to the number of plots in the empirical data, and the expected abundance of each species in each plot equal to its mean regional density (total abundance/regional area). We then estimated (uncorrected) species richness, $S_u$, from each of the samples by fitting Fisher's α to the sampled data and applying the formula $S_u = \alpha \times \ln\left(1 + \frac{N}{\alpha}\right)$. From each of the samples we also derived continental-scale uncorrected SADs (see above), from which the number of hyperdominants, $H_u$, and percentage hyperdominance, $P_u$, could be directly calculated, via analytical expansion of the log series using the fitted values of $\alpha$ and corresponding values of $S_u$. We then regressed the known values of $S_k$, $H_k$ and $P_k$ from the simulated SADs against the estimated (uncorrected) values $S_u$, $H_u$ and $P_u$ from the samples drawn with conspecific aggregation across all 250 simulations—that is, fit linear models of the form $A_k = m \times A_u + c$ for $A = S, H, P$. This same procedure was also applied to the number and proportion of dominants.

Across all three regional datasets, the above procedure outlined a linear relationship with low variance between known values of species richness, number of dominants and hyperdominants, and percentage hyperdominance and dominance, and values thereof estimated from sampling with conspecific aggregation (Extended Data Fig. 5). Thus, constant terms with low variance were readily applicable to correct for bias in the point estimates of species richness, number of dominants/hyperdominants, and percentage hyperdominance/dominance, derived from the empirical Africa, Amazonia, and Southeast Asia data. To capture uncertainty around each bias-corrected point estimate, prediction intervals (PI) were derived as PI = $\mu + 1.96 \times \sigma_{PI}$, where $\mu$ is the predicted mean value of the point estimate according to the linear regression, and $\sigma_{PI}$ is the PI standard error, calculated as $\sigma_{PI} = \sqrt{\sigma^2 + \sigma_R^2}$, where $\sigma$ is the standard error of predicted means and $\sigma_R$ is the residual s.d. (and 1.96 is the 0.05 quantile of a $t$-distribution).

### Reporting summary

Further information on research design is available in the Nature Portfolio Reporting Summary linked to this article.

### Data availability

The species abundance data that support the findings of this study are available from https://doi.org/10.6084/m9.figshare.21670883 (formatting notes: a column for each species, rows for each plot, entries are the number of trees ≥10 cm diameter of each species in each plot). WorldClim[69] bioclimatic data are available from https://www.worldclim.org/data/bioclim.html.

### Code availability

R code (version 4.3.1) to run the analyses and produce the figures and tables is available from https://github.com/declancooper/CommonSpecies2022.git.

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

**Acknowledgements** D.L.M.C. was supported by the London Natural Environmental Research Council Doctoral Training Partnership grant (grant no. NE/L002485/1). This paper developed from analysing data from the African Tropical Rainforest Observatory Network (AfriTRON), curated at ForestPlots.net. AfriTRON has been supported by numerous people and grants since its inception. We sincerely thank the people of the many villages and local communities who welcomed our field teams and without whose support this work would not have been possible. Grants that have funded the AfriTRON network, including data in this paper, are a European Research Council Advanced Grant (T-FORCES; 291585; Tropical Forests in the Changing Earth System), a NERC standard grant (NER/A/S/2000/01002), a Royal Society University Research Fellowship to S.L.L., a NERC New Investigators Grant to S.L.L., a Philip Leverhulme Award to S.L.L., a European Union FP7 grant (GEOCARBON; 283080), Leverhulme Program grant (Valuing the Arc); a NERC Consortium Grant (TROBIT; NE/D005590/), NERC Large Grant (CongoPeat; NE/R016860/1) the Gordon and Betty Moore Foundation the David and Lucile Packard Foundation, the Centre for International Forestry Research (CIFOR), and Gabon's National Parks Agency (ANPN). This paper was supported by ForestPlots.net approved Research Project 81, 'Comparative Ecology of African Tropical Forests'. The development of ForestPlots.net and data curation has been funded by several grants, including NE/B503384/1, NE/N012542/1, ERC Advanced Grant 291585—'T-FORCES', NE/F005806/1, NERC New Investigators Awards, the Gordon and Betty Moore Foundation, a Royal Society University Research Fellowship and a Leverhulme Trust Research Fellowship. Fieldwork in the Democratic Republic of the Congo (Yangambi and Yoko sites) was funded by the Belgian Science Policy Office BELSPO (SD/AR/01A/COBIMFO, BR/132/A1/AFRIFORD, BR/143/A3/HERBAXYLAREDD, FED-tWIN2019-prf-075/CongoFORCE, EF/211/TREE4FLUX); by the Flemish Interuniversity Council VLIR-UOS (CD2018TEA459A103, FORMONCO II); by L'Académie de recherche et d'enseignement supérieur ARES (AFORCO project) and by the European Union through the FORETS project (Formation, Recherche, Environnement dans la TShopo) supported by the XIth European Development Fund. EMV was supported by fellowship from the CNPq (Grant 308543/2021-1). RAPELD plots in Brazil were supported by the Program for Biodiversity Research (PPBio) and the National Institute for Amazonian Biodiversity (INCT-CENBAM). BGL post-doc grant no. 2019/03379-4, São Paulo Research Foundation (FAPESP). D.A.C. was supported by the CCI Collaborative fund. Plots in Mato Grosso, Brazil, were supported by the National Council for Scientific and Technological Development (CNPq), PELD-TRAN 441244/2016-5 and 441572/2020-0, and Mato Grosso State Research Support Foundation (FAPEMAT)—0346321/2021. We thank E. Chezeaux, R. Condit, W. J. Eggeling, R. M. Ewers, O. J. Hardy, P. Jeanmart, K. L. Khoon, J. L. Lloyd, A. Marjokorpi, W. Marthy, H. Ntahobavuka, D. Paget, J. T. A. Proctor, R. P. Salomão, P. Saner, S. Tan, C. O. Webb, H. Woell and N. Zweifel for contributing forest inventory data. We thank numerous field assistants for their invaluable contributions to the collection of forest inventory data, including A. Nkwasibwe, ITFC field assistant.

**Author contributions** D.L.M.C. and S.L.L. conceived and developed the study. D.L.M.C. performed the analysis with M.J.P.S. and P.I.P. and input from S.L.L. D.L.M.C., P.I.P., G.C.P., A.L. and M.J.P.S. developed tools to support the analysis. D.L.M.C. and S.L.L. wrote the manuscript with significant input from M.J.P.S., R.G.P. and M.I.D. S.L.L., B.S. and C.E.N.E. curated the AfriTRON forest plot data. N.B., P.P. and G.D. curated the Central African Plot Network forest plot data. H.T.S. curated the ATDN forest plot data. F.S. curated the Slik et al.[4,25] Southeast Asia forest plot data. S.L.L. and O.L.P. curated the T-FORCES Southeast Asia carbon monitoring network. O.L.P., S.L.L., T.R.B., B.S, B.S.M., C.E., E.H., L.Q., A.L. and G.P. provided ForestPlots.net pan-tropical data management. All co-authors contributed data, reviewed, approved and had the opportunity to comment on the manuscript.

**Competing interests** The authors declare no competing interests.

**Additional information**
**Correspondence and requests for materials** should be addressed to Declan L. M. Cooper or Simon L. Lewis.

**a**

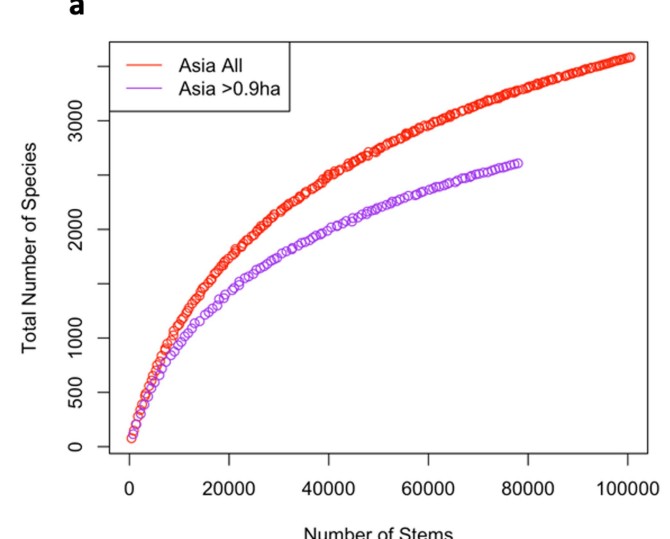

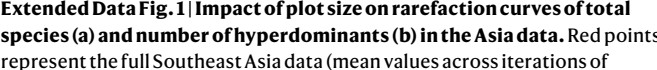

**b**

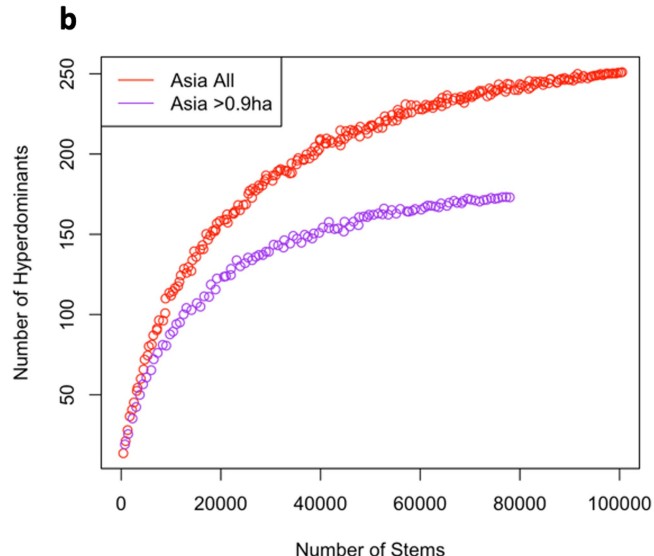

**Extended Data Fig. 1 | Impact of plot size on rarefaction curves of total species (a) and number of hyperdominants (b) in the Asia data.** Red points represent the full Southeast Asia data (mean values across iterations of subsamples), including all plot sizes (mean plot size: 0.877 ha, median plot size: 0.5 ha); Purple points represent the Southeast Asia data restricted to plots ≥0.9 ha (mean plot size: 1.59 ha, median plot size: 1 ha).

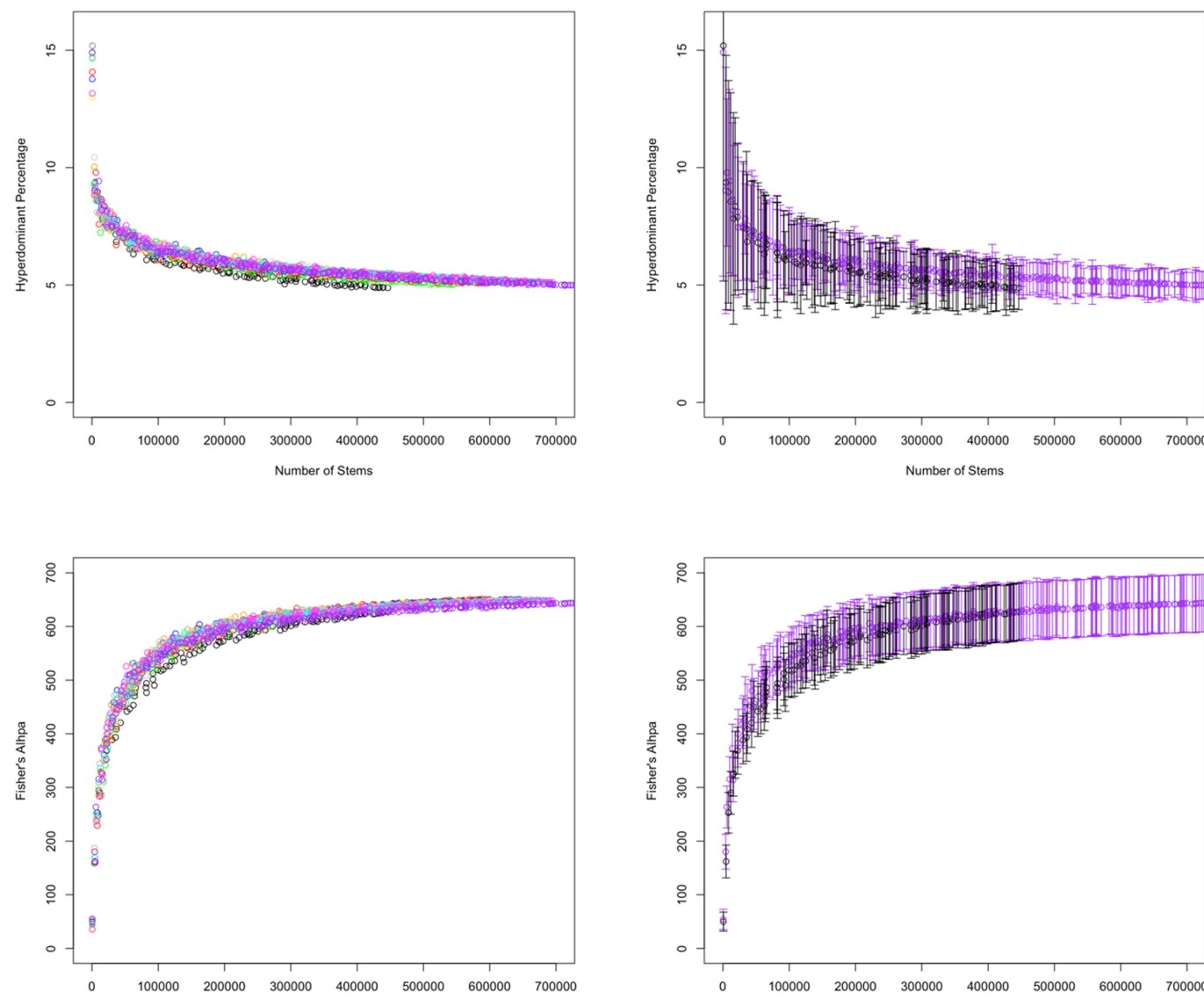

**Extended Data Fig. 2 | Impact of spatial clustering of plots on rarefaction curves of hyperdominant percentage (first row) and Fisher's Alpha (second row) in the Amazonia data.** Purple points and confidence intervals represent the full data; black points and confidence intervals represent a subset of the data in which one plot is sampled from each spatial cluster of plots; other coloured points represent subsets of the data in which 2,3,4,…,10 plots (or the total number of plots in the cluster) are sampled from each spatial cluster of plots. Points give the mean values across iterations of subsamples. Confidence intervals are derived via the standard deviation across iterations of subsamples taken with replacement at each sampling point. Note that although resampling for rarefaction was done by subsampling tree inventory plots, the curves are re-plotted with an x-axis of number of stems.

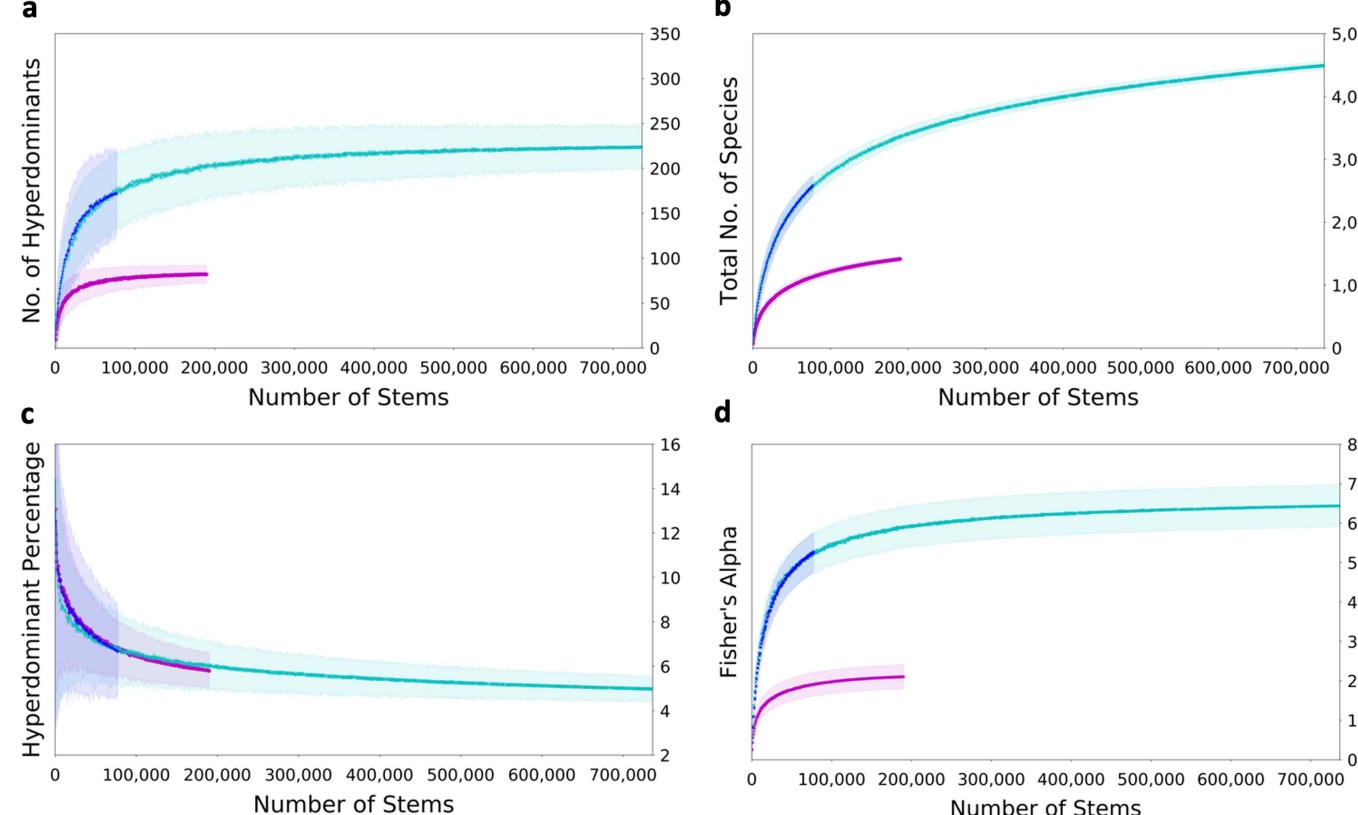

**Extended Data Fig. 3 | Complete rarefaction curves showing the effect of increasing sampling on the number of hyperdominants (a), total species (b), hyperdominant percentage (c), and fitted values of Fisher's α (d).** In tropical Africa (magenta), Amazonia (cyan), Southeast Asia (blue). Markers represent rarefied points (mean values across iterations of subsamples); shaded areas represent confidence intervals (CIs). Confidence intervals are derived via the standard deviation across iterations of subsamples taken with replacement at each sampling point. Note that although resampling for rarefaction was done by subsampling tree inventory plots, the curves are re-plotted with an x-axis of number of stems.

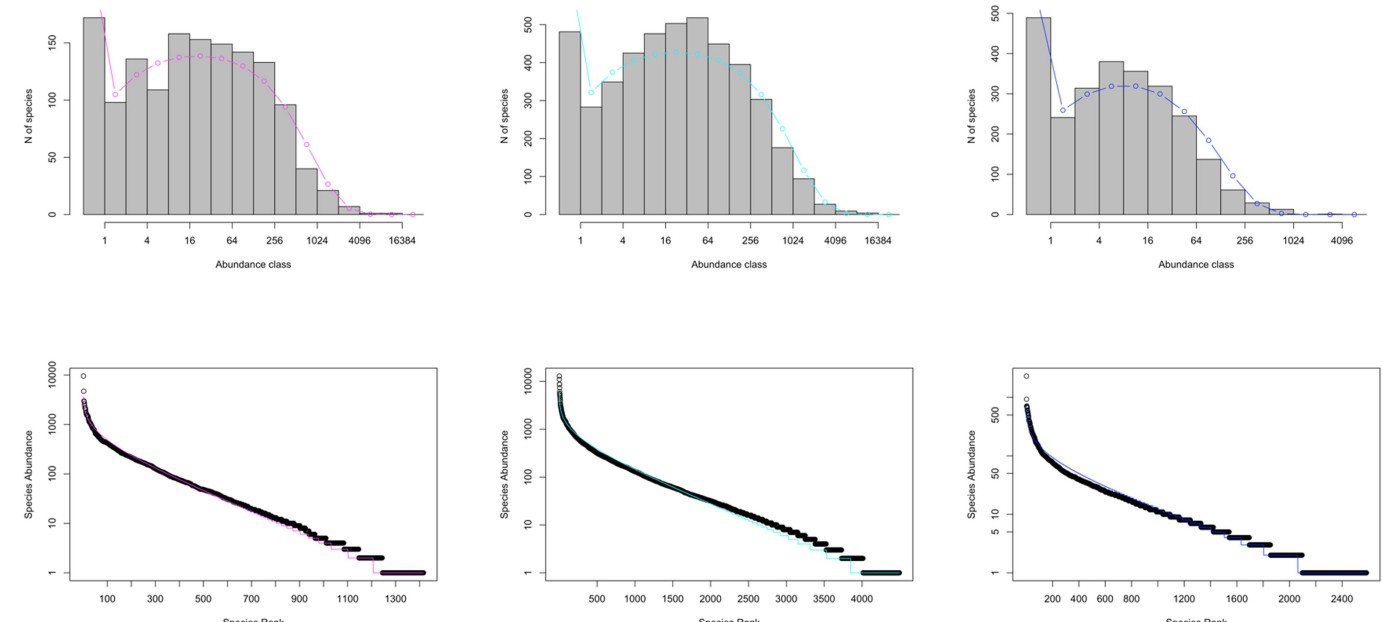

**Extended Data Fig. 4 | Preston plots (top row) and rank abundance distributions (bottom row) showing the empirical species abundance distributions for Africa (left) Amazonia (middle) and Southeast Asia (right) with log series fits overlaid.** Histogram bars display the empirical species abundance distributions as Preston plots (top row); black markers show the empirical species abundance distributions as rank abundance distributions (bottom row); overlaid points and lines show log series fits to empirical species abundance distributions in Africa (magenta), Amazonia (cyan), and Southeast Asia (blue).

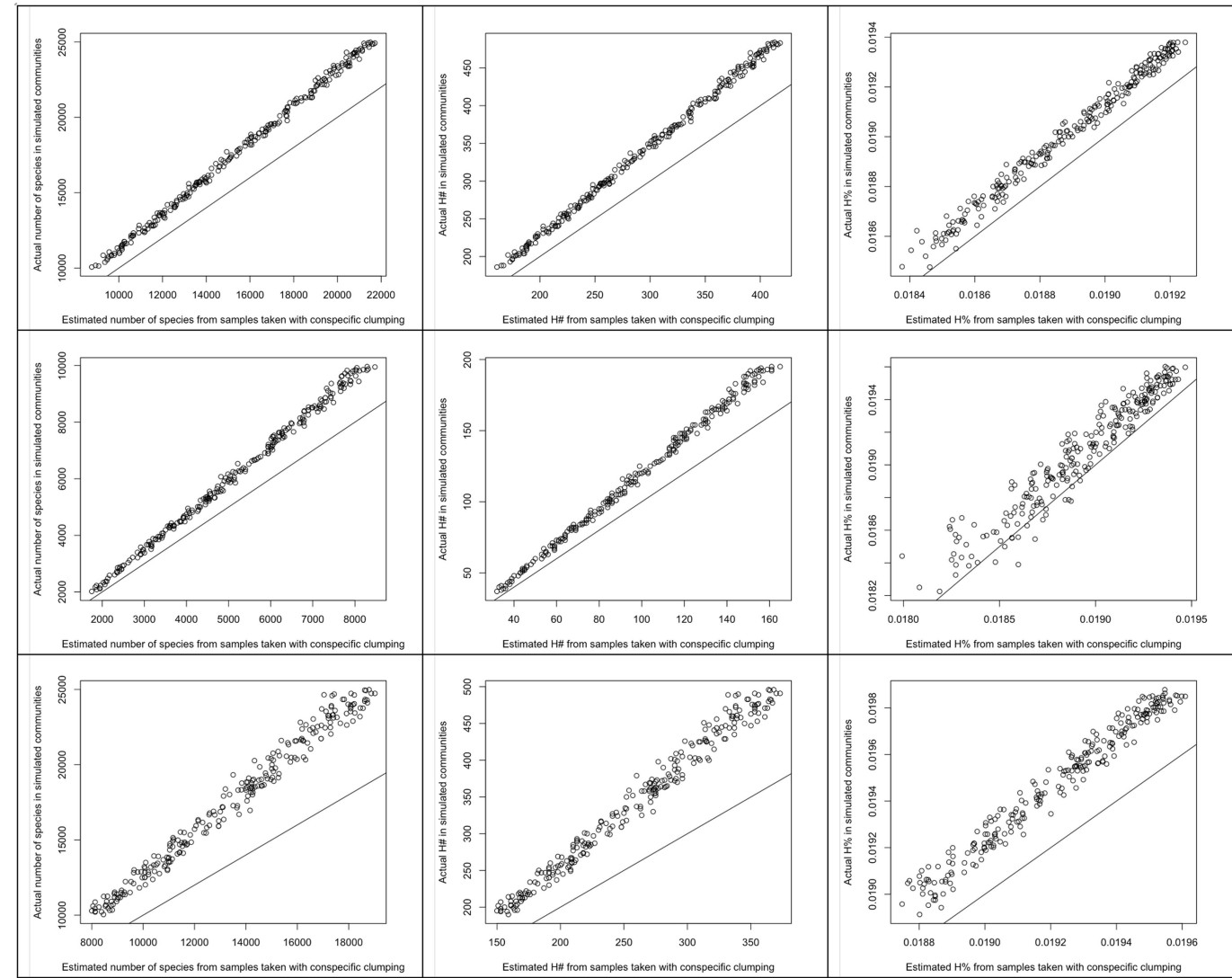

**Extended Data Fig. 5 | Bias correction of estimates of species richness (first column), number of hyperdominants (second column), percentage hyperdominance (third column) for the Amazonia (first row), Africa (second row) and Southeast Asia (third row) datasets.** X-axes show estimated values derived from samples of the simulated communities taken with conspecific aggregation, Y-axes show true values of the simulated communities. Points show estimated true values for each of the 250 simulated communities. 1:1 equivalence shown by straight line in each plot. For number of hyperdominants and total species plots, simulated communities containing 100 to 25,000 species in Amazonia and Southeast Asia, 100 to 10,000 species in Africa are shown. For percentage hyperdominance, simulated communities containing 10,000 to 25,000 species in Amazonia and Southeast Asia, 2,000 to 10,000 species in Africa are shown.

**Extended Data Table 1 | Empirical summary statistics and hyperdominance results for tree species data in Africa, Amazonia, and Southeast Asia**

| | Plots | Stems | %ID | H# | TS | H% | α |
|---|---|---|---|---|---|---|---|
| Africa | 368 | 189,948 | 92.7 | 82 | 1,416 | 5.79 | 210 |
| Amazonia | 1,097 | 736,270 | 93.7 | 224 | 4,492 | 4.99 | 644 |
| Southeast Asia | 103 | 77,587 | 91.6 | 172 | 2,585 | 6.65 | 526 |

#H = Number of hyperdominants, TS = Total Species, H% = Percentage hyperdominance, α = Fisher's α, Stems = Total number of stems, Plots = Total number of plots, %ID = Percentage of stems identified to the species level.

**Extended Data Table 2 | Rarefied minimum number of species required to account for 10%, 20%, …, 90% of trees in the Africa, Amazonia, and Southeast Asia data, resampled to the common sample size of the Asia dataset (77,587 stems)**

| | 10% | 20% | 30% | 40% | 50% | 60% | 70% | 80% | 90% |
|---|---|---|---|---|---|---|---|---|---|
| Africa | 4 [2,7] | 13 [8,17] | 26 [19,33] | 46 [36,57] | 77 [62,92] | 121 [102,139] | 186 [163,209] | 296 [264,328] | 592 [494,689] |
| Amazonia | 8 [4,13] | 28 [17,38] | 60 [42,77] | 106 [79,133] | 174 [134,215] | 276 [214,337] | 434 [339,528] | 709 [553,864] | 1413 [1029,1797] |
| Southeast Asia | 9 [5,13] | 26 [17,35] | 54 [36,72] | 98 [66,130] | 172 [125,219] | 285 [220,350] | 468 [380,556] | 790 [670,910] | 1778 [1427,2129] |

Percentage headings represent the different dominance thresholds. Confidence intervals are derived from the standard deviation across iterations of subsamples taken with replacement at the sample size of the Asia dataset.

**Extended Data Table 3 | Rarefied proportion of total species required to account for 10%, 20%, ..., 90% of trees in the Africa, Amazonia, and Southeast Asia data, resampled to the common sample size of the Asia dataset (77,587 stems)**

| | 10% | 20% | 30% | 40% | 50% | 60% | 70% | 80% | 90% |
|---|---|---|---|---|---|---|---|---|---|
| Africa | 0.36 [0.12,0.61] | 1.11 [0.71,1.51] | 2.28 [1.64,2.93] | 4.09 [3.1,5.07] | 6.79 [5.39,8.20] | 10.67 [8.89,12.44] | 16.47 [14.28,18.65] | 26.18 [23.22,29.15] | 52.28 [43.00,61.56] |
| Amazonia | 0.33 [0.15,0.51] | 1.09 [0.67,1.50] | 2.32 [1.63,3.01] | 4.14 [3.09,5.19] | 6.80 [5.24,8.36] | 10.74 [8.40,13.09] | 16.90 [13.30,20.50] | 27.63 [21.68,33.58] | 55.06 [40.37,69.76] |
| Southeast Asia | 0.35 [0.19,0.51] | 1.01 [0.61,1.41] | 2.09 [1.27,2.91] | 3.79 [2.38,5.20] | 6.65 [4.59,8.71] | 11.03 [8.24,13.82] | 18.10 [14.47,21.73] | 30.56 [25.78,35.34] | 68.78 [52.50,84.97] |

Percentage headings represent the different dominance thresholds. Confidence intervals are derived from the standard deviation across iterations of subsamples taken with replacement at the sample size of the Asia dataset.

**Extended Data Table 4 | Extrapolated minimum number of species required to account for 10%, 20%, ..., 90% of trees in Africa, Amazonia, Southeast Asia, and the cross-regional total at the regional scale**

|  | 10% | 20% | 30% | 40% | 50% | 60% | 70% | 80% | 90% |
|---|---|---|---|---|---|---|---|---|---|
| Africa | 7 [4,10] | 22 [19,25] | 41 [38,44] | 69 [66,71] | 104 [101,107] | 154 [151,157] | 228 [225,231] | 354 [351,357] | 713 [709,717] |
| Amazonia | 20 [16,25] | 62 [58,66] | 120 [116,124] | 196 [192,201] | 299 [295,304] | 443 [438,447] | 651 [647,656] | 1000 [995,1005] | 1892 [1886,1899] |
| Southeast Asia | 12 [1,23] | 51 [40,62] | 106 [95,116] | 179 [168,189] | 278 [268,289] | 417 [406,427] | 625 [614,636] | 988 [977,1000] | 2243 [2225,2262] |
| Total | 39 [21,58] | 135 [117,153] | 267 [249,284] | 444 [426,461] | 681 [664,700] | 1014 [995,1031] | 1504 [1486,1523] | 2342 [2323,2362] | 4848 [4820,4878] |

Percentage headings represent the different dominance thresholds. Prediction intervals combine uncertainty from the standard error of predicted means and the residual standard deviation of the regression of the bias correction fit. 'Total' minimum number of species required to account for 10%–90% of trees across all of the regions are calculated as the sum of the number of hyperdominants across the three major tropical forest regions.

**Extended Data Table 5 | Extrapolated proportion of total species required to account for 10%, 20%, 30%, …, 90% of trees in Africa, Amazonia, Southeast Asia, and the cross-regional total at the regional scale**

|  | 10% | 20% | 30% | 40% | 50% | 60% | 70% | 80% | 90% |
|---|---|---|---|---|---|---|---|---|---|
| Africa | 0.16 | 0.48 | 0.90 | 1.48 | 2.23 | 3.31 | 4.90 | 7.60 | 15.29 |
| Amazonia | 0.14 | 0.44 | 0.86 | 1.41 | 2.16 | 3.21 | 4.72 | 7.25 | 13.73 |
| Southeast Asia | 0.14 | 0.46 | 0.91 | 1.5 | 2.32 | 3.45 | 5.14 | 8.10 | 18.34 |
| Total | 0.13 | 0.44 | 0.88 | 1.46 | 2.24 | 3.33 | 4.94 | 7.70 | 15.93 |

Percentage headings represent the different dominance thresholds. 'Total' minimum proportion of total species required to account for 10%–90% of trees across all of the regions are calculated as the proportion between the sum of the number of hyperdominants and the sum of total species across the three major tropical forest regions.

# Reporting Summary

## Statistics

For all statistical analyses, confirm that the following items are present in the figure legend, table legend, main text, or Methods section.

| n/a | Confirmed | |
|---|---|---|
| ☐ | ☒ | The exact sample size (*n*) for each experimental group/condition, given as a discrete number and unit of measurement |
| ☐ | ☒ | A statement on whether measurements were taken from distinct samples or whether the same sample was measured repeatedly |
| ☒ | ☐ | The statistical test(s) used AND whether they are one- or two-sided<br>*Only common tests should be described solely by name; describe more complex techniques in the Methods section.* |
| ☐ | ☒ | A description of all covariates tested |
| ☐ | ☒ | A description of any assumptions or corrections, such as tests of normality and adjustment for multiple comparisons |
| ☐ | ☒ | A full description of the statistical parameters including central tendency (e.g. means) or other basic estimates (e.g. regression coefficient) AND variation (e.g. standard deviation) or associated estimates of uncertainty (e.g. confidence intervals) |
| ☒ | ☐ | For null hypothesis testing, the test statistic (e.g. *F*, *t*, *r*) with confidence intervals, effect sizes, degrees of freedom and *P* value noted<br>*Give P values as exact values whenever suitable.* |
| ☒ | ☐ | For Bayesian analysis, information on the choice of priors and Markov chain Monte Carlo settings |
| ☒ | ☐ | For hierarchical and complex designs, identification of the appropriate level for tests and full reporting of outcomes |
| ☒ | ☐ | Estimates of effect sizes (e.g. Cohen's *d*, Pearson's *r*), indicating how they were calculated |

*Our web collection on statistics for biologists contains articles on many of the points above.*

## Software and code

Policy information about availability of computer code

| Data collection | No software was used for data collection. |
|---|---|
| Data analysis | The custom R code (version 4.3.1) used to run the analyses and produce the figures is available on GitHub with the identifier https://github.com/declancooper/CommonSpecies2022.git |

For manuscripts utilizing custom algorithms or software that are central to the research but not yet described in published literature, software must be made available to editors and reviewers. We strongly encourage code deposition in a community repository (e.g. GitHub). See the Nature Portfolio guidelines for submitting code & software for further information.

## Data

Policy information about availability of data

All manuscripts must include a data availability statement. This statement should provide the following information, where applicable:
- Accession codes, unique identifiers, or web links for publicly available datasets
- A description of any restrictions on data availability
- For clinical datasets or third party data, please ensure that the statement adheres to our policy

The species abundance data that support the findings of this study are available from https://doi.org/10.6084/m9.figshare.21670883. WorldClim bioclimatic data are available from https://www.worldclim.org/data/bioclim.html.

## Human research participants

Policy information about studies involving human research participants and Sex and Gender in Research.

| | |
|---|---|
| Reporting on sex and gender | NA |
| Population characteristics | NA |
| Recruitment | NA |
| Ethics oversight | NA |

Note that full information on the approval of the study protocol must also be provided in the manuscript.

# Field-specific reporting

Please select the one below that is the best fit for your research. If you are not sure, read the appropriate sections before making your selection.

☒ Life sciences      ☐ Behavioural & social sciences      ☐ Ecological, evolutionary & environmental sciences

For a reference copy of the document with all sections, see nature.com/documents/nr-reporting-summary-flat.pdf

# Life sciences study design

All studies must disclose on these points even when the disclosure is negative.

| | |
|---|---|
| Sample size | No sample size calculation was performed. We selected all available plots meeting the following criteria: forest inventory plots ≥0.2 ha in size, situated in structurally intact (no logging or fire), closed canopy (not dry forest or savanna) tropical forest, with enumeration of all stems ≥ 10 cm diameter, in which ≥ 80% of stems are identified to the species level. These criteria allow direct comparisons to be made with results from previous studies investigating common species abundances in Amazonia. The major tropical forest regions of Amazonia, Africa, and Southeast Asia are adequately represented. This is the largest dataset of repeatedly measured plots ever used to calculate long-term trends in African forest carbon dynamics. |
| Data exclusions | Plots not meeting the following criteria were excluded from analysis: forest inventory plots ≥0.2 ha in size, situated in structurally intact (no logging or fire), closed canopy (not dry forest or savanna) tropical forest, with enumeration of all stems ≥ 10 cm diameter, in which ≥ 80% of stems are identified to the species level. These conditions were pre-established in line with previous studies. |
| Replication | Replication of repeated random subsampling 200 times ensured that derived results were stable and reproducible. |
| Randomization | Forest inventory data were partitioned into the the Amazonia, Africa, and Southeast Asia study regions by location. In all of the analyses, sampling and sub-sampling by plots and by trees within plots was done completely randomly within regions. |
| Blinding | Blinding is not relevant to this study. |

# Reporting for specific materials, systems and methods

We require information from authors about some types of materials, experimental systems and methods used in many studies. Here, indicate whether each material, system or method listed is relevant to your study. If you are not sure if a list item applies to your research, read the appropriate section before selecting a response.

### Materials & experimental systems

| n/a | Involved in the study |
|---|---|
| ☒ ☐ | Antibodies |
| ☒ ☐ | Eukaryotic cell lines |
| ☒ ☐ | Palaeontology and archaeology |
| ☒ ☐ | Animals and other organisms |
| ☒ ☐ | Clinical data |
| ☒ ☐ | Dual use research of concern |

### Methods

| n/a | Involved in the study |
|---|---|
| ☒ ☐ | ChIP-seq |
| ☒ ☐ | Flow cytometry |
| ☒ ☐ | MRI-based neuroimaging |

