## [Peer Review File · Nature]

Manuscript Title: Consistent Patterns of Common Species Across Tropical Tree Communities

Reviewer Comments & Author Rebuttals

Reviewer Reports on the Initial Version:

Referees' comments:

Referee #1 (Remarks to the Author):

The manuscript submitted by Cooper and colleagues uses a large collection of forest inventory plots across 3 continents to quantify the number and importance of hyper-dominant species. The team uses, primarily, 1 hectare inventory plots and focus on trees 10cm in diameter and higher. The authors conclude that roughly 2% of tree species in the data and, throughout, the tropics are hyper-dominant and account for a very high amount of individuals, and therefore biomass, in these forests. The resulting arguments are that consistent community assembly processes are likely and that we should place more focus on these very common species.

From a technical/analytical perspective, I see no major flaws with this work. There can always be more forest plot data and we should be careful with extrapolations, particularly from relatively small plots, but I doubt the results would change dramatically with more data from these regions or others (e.g. Papua NG).

That said, I do not find the results that surprising and they are unlikely to change or improve our understanding of these systems. Indeed, the work largely demonstrates one of the few well-known rules in ecology - an approximately log-normal species abundance distribution (SAD). That is, very few species are very common and very many species are very rare. The Discussion is interesting in this regard as it quotes Darwin, but ignores the very large literature on SADs. In sum, the fact that SADs are repeatable across regions/systems is perhaps one of the few things ecologists would agree upon as likely to be present in data before seeing it.

The argument that repeated shapes to SADs being evidence of consistent assembly mechanisms is an old one and fraught. This spans from broken stick niche partitioning and MacArthur all the way through to Hubbell's neutral model approximating the expected SAD and McGill's rebuttal. Other work from John Harte has shown that realistic SADs can simply arise using statistical mechanics where maximum entropy is constrained by the number of species and individuals. What became clear from this literature is the SAD patterns are challenging to link to a specific process or even easily used for differentiating between stochastic and deterministic processes.

Some have used alternate approaches. For example, Ricklefs and Renner (Science 2012) have argued similarity in what lineages are in the common or rare parts of the SAD across systems is evidence of determinism. This is somewhat more compelling evidence, but already documented.

The final argument that we should focus on common species is interesting, but likely in the eye of

the beholder. We almost certainly should focus on these species if we are keen on understanding biomass/carbon/etc. I do not dispute this. However, many other research topics (e.g. co-existence or community stability) likely require just as much information about rare species. Enquist and colleagues in 2019 in *Science Advances* working on the "commonness of rarity" make arguments akin to this. Indeed, their work seems to argue for the opposite of what is posed in the present MS. In the end, neither perspective is right in all instances, of course, and I worry one takeaway here is we should only adhere to one of these perspectives.

In sum, the work is technically robust, but it is demonstrating one of the most well-known ecological rules. It is jarring to read a MS that suggests what is found is exceptional and not what should be expected. The tentative inference that universal mechanisms are at play due to the repeated SAD shape is known, in the community ecology literature, to be fraught with problems and the argument we should focus on common species is valid for a subset of pursuits, but is a damaging argument for many other important sub-disciplines.

Referee #2 (Remarks to the Author):

Using an extraordinary database generated from the census of more than one million trees (DBH \geq 10 cm) in thousands of plots established across different tropical regions of the Americas, Africa, and Asia. Cooper et al. find that in the world's moist tropical and subtropical forests, only ~2% of the species cover 50% of the total trees. The paper also provides an estimate and a list of the most common tree species (~1,000) that could be found globally in these forest types. This is an exceptional and novel result in the literature on tropical forests. These results represent a very important contribution to advance in the understanding of the biology of tropical forests, and I believe that the consistency of the results opens up very important avenues of research in the fields of universal knowledge as well as application in topics of conservation and management of the most biodiverse ecosystems on the planet. In this sense, I believe that the great collaborative work carried out to arrive at the reported results represents a significant contribution to the search for general patterns in the field of forest biology in general and tropical ecology in particular.

The paper reads easily and has a good balance between the different sections. Reviewing the paper, I detect some observations that I consider should be further addressed to strengthen the significance of the findings.

1. The authors indicate that identifying common species can help to advance faster in understanding the ecology of hyperdiverse tropical forests and in the responses of these forests to changes in the planetary system. However, I did not find any part of the paper that discusses how studying the biology/ecology of common species can help this rapid advance. I suggest including in the discussion a brief argumentation on this issue.

2. It is interesting that the forests of Africa are consistently less species-rich than those of the Americas and Asia. However, this result is not discussed despite the fact that such a difference was a consistent pattern as the one found for the percentage of common species. I suggest that the

discussion address possible reasons (biogeographic, historical human activity, others) why species richness was lower in African forests.

3. It is relevant to find that a quite similar percentage of common species occurs in the different forests analyzed in the three continents. But it's also interesting that 50% of the stems contain ~93% of the species. This data underlies one of the most relevant study issues of the biology of tropical forests: what determines the great diversity of species? I think authors should highlight this result and discuss, based on the existing literature, possible explanations for the great richness of rare species (for example, theories about the coexistence of species in hyperdiverse systems).

4. Also, rare species are extremely important for conserving the greatest biodiversity on the planet. It is important to clarify that it would be a bad strategy only to concentrate efforts on common species, neglecting rare species. It is also necessary to carry out research efforts and, above all, the conservation of rare species. Perhaps the concept of umbrella species can be used, indicating that conserving common species could also conserve rare species.

5. One aspect I suggest discussing is that the common species by abundance may not be the dominant species by biomass. Therefore, estimating the number and percentage of species that cover 50% of the biomass would be relevant. I can imagine that many common species are those that inhabit the understory and that they are not necessarily the ones that contribute a significant fraction of the forest's biomass. Common species by abundance and dominant species by biomass could play different roles in tropical forest ecosystems' structure, dynamics, and function.

6. The authors indicate that a similar percentage of common species was found in the forests of the three continents. This implies the existence of a fundamental and general assembling rule of tree communities. However, the paper does not discuss what mechanisms may be underlying this rule. In the literature, different deterministic and neutral mechanisms have been proposed that could be used to expose ideas that help to understand if this is a statistical rule (for example, it has been found that ordering the words of a book by their frequency of appearance can generate a log-normal distribution) or it has tractable ecological hypotheses to explain the similarity in the low percentage of common species.

7. To go beyond the finding of the low percentage of common species prevalent throughout the planet's tropical forests, the authors could discuss (perhaps supported by data) the role that common species can play in the maintenance of forest biodiversity. For example, one could find out the frequency of common species pollinated by animals and the frequency of common species producing fleshy fruits eaten by frugivorous and seed-dispersing animals. How common species contribute to the structure of tree communities can also be discussed, for example, by accounting for the frequency of species that are in the understory and those that occupy the canopy. Finally, it would be important to highlight the role of common species as a source of ecosystem services. For example, in addition to functioning as carbon sinks and storage, common species can be a source of current or potential forest products for societies. This issue could be discussed shortly.

Technical aspects

It would be useful to provide data on climate (for example, variation in total annual precipitation between sites) and elevation above sea level of the study locations in the supplementary information. These data can help the reader to understand the similarities/differences between the forest types that were included in the study beyond the information indicating that tropical forests that fall into the “Tropical and Subtropical Moist Broad Leaf Forest” biome delimited by the ecoregions map of the World Wildlife Fund were included.

I believe that a brief discussion should be included on the precision/confidence of the method used to estimate the global number of trees in tropical forests, considering that in some cases, the estimation for some regions was based on data from a single plot. This discussion may be indicated in the supplementary information.

Referee #3 (Remarks to the Author):

General comments

The underlying idea of the manuscript is that we could understand the (dynamics of) tropical forests and their response to environmental change if only a small proportion of species would make up the majority of the larger tree individuals (i.e., “hyperdominant species”), for which it would be feasible to collect data on their natural history. However, the order of magnitude of hyperdominant species and their differences between tropical regions is not known.

The study compiles a large data set of small plots in tropical forests across Africa, Amazonia and Southeast Asia to assess the proportion of hyperdominant species in these three regions, to extrapolate them to the entire closed-canopy forests of the tropics, and to derive a list of species in their data set that are likely to be hyperdominant.

The results show a striking similarity in the proportion of hyperdominant species among the 3 regions and extrapolation suggests that a manageably small number of tree species (around 1000) may account for half of Earth’s trees (with diameter >10 cm) found in the closed-canopy tropical forest biome. On overall, the underlying idea and the results of the study are interesting and highly relevant for applied and theoretical ecologists, but will also appeal to a broader audience. The striking similarity in the patterns of species abundances in tropical forests among continents is also likely to stimulate theoretical progress in explain this pattern.

Extrapolating the number of species and their abundances from a relatively small number of plots to the entire Amazonian region or even the entire tropics is a complex and very much debated undertaking. As a consequence, a number of recent studies have contributed to this issue by testing different species abundance distributions and methods to remove bias in the extrapolation. The manuscript rests on a recent method by ter Steege et al. (2020) that derived biased-corrected richness estimates for the Amazonian tree flora by recognizing that conspecific spatial aggregation at the sampling scale, which is a serious source of bias of parametric estimates of species richness,

increases with rarity.

I understand the difficulties to derive such estimates, but also the need to put the available information together to make educated guesses. Nevertheless I feel a bit uneasy regarding some assumptions of the methods. Although ter Steege et al. (2020) addressed the issue of non-random species distributions, the potential effect of non-random sampling (i.e., aggregated location of the sampling plots) is not addressed. To assess the robustness of the results I recommend a simulation study where realistic large spatially-explicit tree maps (which provide also known species abundance distributions) are generated by a process based model and sampled with spatial placement of the sampling plots similar to the actual one. A dynamic model with spatially-explicit dispersal is required to produce realistic distance-dependent correlations in the species composition among plots that can lead to bias if plots are aggregated. Doing such an analysis for sufficiently large tree maps should be feasible using the software of Thompson et al. (2020: DOI: 10.1111/2041-210X.13451).

Specific comments

Unfortunately, the pdf did not contain line and page numbers, I therefore refer to concrete places in the manuscript using the pdf page numbers, but only to the paragraph level.

page 8p2

Be concrete, what are the “fundamental rules” or “fundamental processes”? Species abundance distributions have been intensely studied with theoretical models (e.g., neutral theory and extensions that provide mechanistic interpretations of the parameters of the LogSeries). This unspecified term is overused since it appears also at page 7p1, page 8p2, page 10p2, page 16p1, page 16p2, and page 17p1.

page 8p2, page 17p2

This is true, however, non-hyperdominant species may comprise a reservoir of species that become more important (dominant) and function as a buffer if environmental conditions change. See also the diversity-functioning debate. I think this should be discussed.

page 9p2

How does “trunk diameter” relate to dbh?

page 9p3

I miss an explanation why including smaller plots inflates the total number of species per number of stems (Extended Data Fig. 1). I guess that this is an effect of species aggregation and beta diversity (distance decay of similarity) where two or more smaller but spatially separated plots host more species than one larger plot of their joined size. How do the remaining differences in plot sizes influence results? The simulation tests (e.g., ter Steege et al. 2020) of the method used 1-ha plots, were most sampling plots 1-ha plots?

Given the sensitivity of results to this spatial issue, it is likely that the spatial arrangement of the plots, which is highly non-random, interacts with species aggregation and beta diversity in biasing the results. I therefore recommend that the authors test their analysis against realistic simulated

spatially-explicit tree maps, e.g., using the software of Thompson et al. (2020).

Page 10p1+2

Indeed, the pan-tropical consistency in the proportion of hyperdominant species is a striking result!

Page 12p1

Extended data Fig. 3 was not provided, I guess you mean Extended Data Figure 4? Should you here not also provide the estimated number of trees in each study region (that are later given at page 19)? The estimate of 299 hyperdominant species in Amazonia match the estimate of Tovo et al. (2017).

Page 15p1

Using simulated tree maps could also help to test the methodology to detect hyperdominant species.

Page 16p1+2

Here you should go deeper and discuss theories that may explain the fundamental rules of tree community assembly that may lead to consistent patterns of species abundance.

Page 18p2+3+4

You should provide more detailed information on the size distribution of the plots. Tests of your methodology rest on 1-ha plots.

Page 20p1

The spatial pattern of plots is also likely to introduce some bias into your estimates.

Page 21p3

Please mention that the approach considered species-specific aggregation at the plot scale depending on species density.

Page 23p2

You mean Extended Data Fig. 5?

Author Rebuttals to Initial Comments:

DLMC&SLL: We thank the reviewers of the paper. Rev#1 assessed the manuscript as technically robust, but unlike Rev#2 and Rev #3, they did not find our results surprising. Rev#2 assessed the manuscript as “a very important contribution to advance in the understanding of the biology of tropical forests”. Rev#3 said, “the pan-tropical consistency in the proportion of hyperdominant species is a striking result”, and that “the underlying idea and the results of the study are interesting and highly relevant for applied and theoretical ecologists, but will also appeal to a broader audience”.

In response, we positively address each reviewer point-by-point below. There are no changes to the results or conclusions, but there are three changes to the manuscript in terms of presentation:

1. We have moved the point that a broad ‘common is rare, rare is common’ pattern has been found in many taxa to the introduction to build on what is known, re-framing the opening of the paper, including more care to distinguish between our two hypothesised outcomes, (i) important differences in commonness among continents, but within the expected broad pattern of many more rare species than common species, and (ii) highly consistent patterns. This addresses a key concern of Rev#1.
2. We now discuss potential mechanisms governing community assembly and what we can and cannot deduce from our results, citing the relevant literature, asked for by all reviewers.
3. We address the concern that focusing on common species may have the unintended consequence that we do not focus on rare species that are important for conservation and other scientific questions (e.g. coexistence of species). We now address this clearly in the abstract as well as in the discussion to avoid this. We emphasise that focusing on common species is a complementary approach to understanding tropical forests better, but should in no way detract from conservation efforts for rare species, nor scientific questions relating to rare species. This addresses concerns of Rev#1 and Rev#2.

There are two additional analyses, one addressing the non-random sampling of the plots across the continent, suggested by Rev#3, which does not have an impact on the presented results or interpretation. This new analysis is presented in Extended Data Fig. 2. The second analysis, also suggested by Rev#3, limits plot selection to only those of 1ha plots only, which shows the same consistency across the continents (reported in the methods).

The comments by all three reviewers have made the manuscript clearer and more robust and are much appreciated. Our point-by-point responses to the reviewer comments are below.

Referees' comments:

Referee #1 (Remarks to the Author):

The manuscript submitted by Cooper and colleagues uses a large collection of forest inventory plots across 3 continents to quantify the number and importance of hyper-dominant species. The team uses, primarily, 1 hectare inventory plots and focus on trees 10cm in diameter and higher. The authors conclude that roughly 2% of tree species in the data and, throughout, the tropics are hyper-dominant and account for a very high amount of individuals, and therefore biomass, in these forests. The resulting arguments are that consistent community assembly processes are likely and that we should place more focus on these very common species.

From a technical/analytical perspective, I see no major flaws with this work. There can always be more forest plot data and we should be careful with extrapolations, particularly from relatively small plots, but I doubt the results would change dramatically with more data from these regions or others (e.g. Papua NG).

DLMC&SLL: Thank you for the positive comment.

That said, I do not find the results that surprising and they are unlikely to change or improve our understanding of these systems. Indeed, the work largely demonstrates one of the few well-known rules in ecology - an approximately log-normal species abundance distribution (SAD). That is, very few species are very common and very many species are very rare. The Discussion is interesting in this regard as it quotes Darwin, but ignores the very large literature on SADs. In sum, the fact that SADs are repeatable across regions/systems is perhaps one of the few things ecologists would agree upon as likely to be present in data before seeing it.

DLMC&SLL: We both agree and disagree with these comments.

First, while we do quote Darwin as sentence 1 of the discussion, we then write in sentence 2: "Our results concur: despite their formidable diversity, the trees in tropical forests fit the 'rare is common, common is rare' pattern³⁰ which has been documented in many other taxa^{31,32}". We did not give sufficient prominence to past work in the previous draft but we did not ignore it – we have now addressed this following valuable reviewer feedback – see below.

Second, we agree that finding a smaller number of common species relative to rare species is not surprising -- the broad shape of SADs is found repeatedly across almost all taxa (and in non-biological complex systems). We disagree in that we -- the manuscript co-authors -- and the two reviewers below find the almost identical patterns of common species (not mere broad similarity of lots of rare compared to common species) across three continents both surprising and of importance. What is new here is the quantification of the patterns and what this reveals about these systems. Specifically, for

the first time we quantify the consistency in patterns between disparate regions and identify and list what are likely the most common tree species of the tropics, a major breakthrough for tropical ecology.

Third, we dispute that these results do not improve our understanding of these systems, given that we (a) show that once we control for sampling and sample size difference we can explain published differences in dominance patterns for the world's most extensive tropical forest region, the Amazon, (b) quantify, for the first time, the number and proportions of hyperdominants for African and Southeast Asian tropical forest regions, (c) show that the dominance patterns for the three tropical continents are near-identical, and (d) provide, for the first time, a list of the most likely identities of the most common species across the continents.

In response, we have now re-framed the introduction of the paper to highlight the past literature at the outset, in the introduction, and more carefully distinguish the two hypothesised outcomes of our three continents 'experiment'. Paragraph 2 (lines 45-59) of the paper now reads:

Our understanding of tropical forests may improve through a focus on the most common tree species. This is a promising avenue given that species abundance distributions showing a modest number of common species and much larger numbers of rare species have been documented across taxa globally¹⁰⁻¹³. Indeed, analyses of tropical forest inventory data from Amazonia have shown that a relatively small number of common species comprise a majority of trees in the region¹⁴⁻¹⁹. However, whether such patterns hold in other tropical forests is unknown, as there have been no comparable analyses for African or Southeast Asian tropical forests. Perhaps, given the substantial differences in total tree species richness²⁰, forest structure²¹, contemporary climate²², and biogeographic and human-occupancy histories²³ among continents, important contrasts in patterns of common species would be expected. Alternatively, if the same processes or mechanisms apply to all tropical forests²⁴, highly consistent patterns may be expected. Crucially, if a tractably modest number of common species do comprise the majority of tropical trees on Earth, this could open new ways of understanding tropical forests via investigating the ecology of the common species.

The argument that repeated shapes to SADs being evidence of consistent assembly mechanisms is an old one and fraught. This spans from broken stick niche partitioning and MacArthur all the way through to Hubbell's neutral model approximating the expected SAD and McGill's rebuttal. Other work from John Harte has shown that realistic SADs can simply arise using statistical mechanics where maximum entropy is constrained by the number of species and individuals. What became clear from this literature is the SAD patterns are challenging to link to a specific process or even easily used for differentiating between stochastic and deterministic processes.

Some have used alternate approaches. For example, Ricklefs and Renner (Science 2012) have argued similarity in what lineages are in the common or rare parts of the SAD across systems is evidence of determinism. This is somewhat more compelling evidence, but already documented.

DLMC&SLL: We agree that it is very difficult to determine the mechanism(s) behind community assembly, so did not attempt to elucidate this in our originally submitted manuscript. Our primary goal is to show the patterns, and whether they are (a) very similar or not, (b) sufficiently dominated by a few species to be of practical use, and (c) make a first attempt at identifying which species are common in the tropics. We left the 'why' as a question for future research. We see that not discussing why in the original manuscript was a mistake on our part, which did not help the readers. We now address this in the discussion, with three new paragraphs to give context, credit to past work, nuance on this fraught issue and the progress that can be made on what we can deduce. At line 312-345 we now write:

Nevertheless, consistent patterns of commonness do not necessarily imply the same causal mechanism(s). The ubiquity of the broad 'rare is common, common is rare' pattern in ecology, which is also found in non-biological complex systems⁴⁵, means inferences as to the cause of this broad pattern are challenging^{24,46}. While combinatoric methods⁴⁶ and models that maximise the entropy of information^{47,48} both produce the ubiquitous 'reverse lazy-J' pattern, empirical observations show fewer common species and more rare species than expected by statistical controls alone⁴⁶. Similarly, neutral models produce the same broad pattern, but produce too few individuals of the most common Amazonian tree species⁴⁹. This suggests that biological mechanisms influence tree community assembly to produce a consistent proportion of common species across continents.

Considering the substantially smaller regional species pool in Africa compared to Amazonia and Southeast Asia, one might expect differing continental patterns of species dominance if evolutionary drivers were the primary mechanism, not the consistent patterns that we find. Similarly, if environmental filtering were a key mechanism, the different contemporary environments, with Africa much drier on average than the other two continents²², and Southeast Asia consisting of scattered island-like areas of forest compared to the contiguous forested region of Amazonia, would also imply differing continental patterns of species dominance, not the near-identical patterns that we find. These constraints limit the potential mechanisms that could apply across our three-continent context.

One potential cross-continental mechanism is dispersal limitation, where the dispersal capabilities of species result in some suitable habitat patches remaining unoccupied. Another mechanism is density- or distance-dependent mortality, which appears widespread across tropical forests⁵⁰. Here,

specialist natural enemies such as pathogens and herbivores reduce seed or juvenile conspecific survival rates near conspecific adults or areas of high juvenile conspecific density, thereby reducing competitive exclusion and contributing to the maintenance of high tree species richness in tropical forests⁵⁰. It is possible that common species have largely evaded density- and/or distance-dependent mortality. Further progress on putative mechanisms can be made, for example, by exploring whether ecological or functional traits differ between common and rare species, and assessing the consistency of any differences among tropical continents⁵¹. While deducing mechanisms is complex, the identification of a tractable number of common species in tropical forests will facilitate progress in understanding of tropical forests beyond species-abundance distributions.

The final argument that we should focus on common species is interesting, but likely in the eye of the beholder. We almost certainly should focus on these species if we are keen on understanding biomass/carbon/etc. I do not dispute this. However, many other research topics (e.g. co-existence or community stability) likely require just as much information about rare species. Enquist and colleagues in 2019 in *Science Advances* working on the "commonness of rarity" make arguments akin to this. Indeed, their work seems to argue for the opposite of what is posed in the present MS. In the end, neither perspective is right in all instances, of course, and I worry one takeaway here is we should only adhere to one of these perspectives.

DLMC&SLL: We certainly agree that neither a focus on rare or common species is right in all instances. The novel aspect of our manuscript, focussing on the common species, should not be to the detriment of also focussing on the rare species. However, we identify and list the most common tree species in the tropics for the first time, allowing a focus on those species. This finding is not about elevating the relative importance of common or rare species in ecology. We now make this broad point clearer in the Abstract, adding a clause to the final line, writing:

Our results should not detract from the importance of understanding and protecting rare species but do open new opportunities to understand the world's most diverse forests, including modelling their response to environmental change, by focusing on the common species that comprise the majority of their trees.

We also finish the paper noting that the rare species are crucial, stating (lines 384-391):

Of course, striving to understand and protect rare and non-hyperdominant species remains crucial, particularly as they face greater extinction risk and may also contribute to the functioning of ecosystems, particularly when more functions^{55,56}, longer timescales⁵⁷, and imposed environmental changes⁵⁶ are considered, and given that the hyperdominants of the future may be rarer today.

Nonetheless, with a complementary grasp of the most common species, mapping, understanding, and modelling of the world's tropical forests will be a much more tractable proposition.

In sum, the work is technically robust, but it is demonstrating one of the most well-known ecological rules. It is jarring to read a MS that suggests what is found is exceptional and not what should be expected. The tentative inference that universal mechanisms are at play due to the repeated SAD shape is known, in the community ecology literature, to be fraught with problems and the argument we should focus on common species is valid for a subset of pursuits, but is a damaging argument for many other important sub-disciplines.

DLMC&SLL: We suspect the difference of opinion here is that of course there are many rare species compared to common species, so an approximate 'lazy J' SAD is *a priori* expected – we think that is now clear in the opening of the manuscript, and with some small edits throughout. The surprising finding is that the patterns of commonness we find are near-identical on three continents, and the practical finding is producing the first list of the common tree species of the world's most biodiverse biome. We hope we have made this clearer in the manuscript and in our comments above. Addressing these points has much improved the manuscript, thank you for the suggestions.

Referee #2 (Remarks to the Author):

Using an extraordinary database generated from the census of more than one million trees (DBH \geq 10 cm) in thousands of plots established across different tropical regions of the Americas, Africa, and Asia. Cooper et al. find that in the world's moist tropical and subtropical forests, only ~2% of the species cover 50% of the total trees. The paper also provides an estimate and a list of the most common tree species (~1,000) that could be found globally in these forest types. This is an exceptional and novel result in the literature on tropical forests. These results represent a very important contribution to advance in the understanding of the biology of tropical forests, and I believe that the consistency of the results opens up very important avenues of research in the fields of universal knowledge as well as application in topics of conservation and management of the most biodiverse ecosystems on the planet. In this sense, I believe that the great collaborative work carried out to arrive at the reported results represents a significant

contribution to the search for general patterns in the field of forest biology in general and tropical ecology in particular.

The paper reads easily and has a good balance between the different sections. Reviewing the paper, I detect some observations that I consider should be further addressed to strengthen the significance of the findings.

DLMC&SLL: Thank you for the positive comments.

1. The authors indicate that identifying common species can help to advance faster in understanding the ecology of hyperdiverse tropical forests and in the responses of these forests to changes in the planetary system. However, I did not find any part of the paper that discusses how studying the biology/ecology of common species can help this rapid advance. I suggest including in the discussion a brief argumentation on this issue.

DLMC&SLL: Thank you for the suggestion, we now elaborate on this in the discussion, at lines 368-373:

Our list of candidate hyperdominants can therefore readily serve new research, including in facilitating targeted autecological data collection to understand their role in providing ecological functions and services. Practically, this species-specific information could enhance tropical forest modelling by focussing on common species instead of relying on 'functional types' or traits, thereby potentially improving predictions of future forest change.

2. It is interesting that the forests of Africa are consistently less species-rich than those of the Americas and Asia. However, this result is not discussed despite the fact that such a difference was a consistent pattern as the one found for the percentage of common species. I suggest that the discussion address possible reasons (biogeographic, historical human activity, others) why species richness was lower in African forests.

DLMC&SLL: This was initially omitted for brevity but we have now added at lines 293-298:

This consistency is all the more striking given relatively lower tree species richness of African tropical forests compared to Amazonian and Southeast Asian forests, likely due to elevated extinction rates in African forests, with evidence of major losses of African species at the Oligocene-Miocene boundary³⁹, and contractions of rainforest area due to drier conditions during repeated glacial-interglacial cycles over the last 2.6 million years⁴⁰.

3. It is relevant to find that a quite similar percentage of common species occurs in the different forests analyzed in the three continents. But it's also interesting that 50% of the stems contain ~93% of the species. This data underlies one of the most relevant study issues of the biology of tropical forests: what determines the great diversity of species? I think authors should highlight this result and discuss, based on the existing literature, possible explanations for the great richness of rare species (for example, theories about the coexistence of species in hyperdiverse systems).

DLMC&SLL: We agree that it is interesting. We now note the vast numbers of rare species in the discussion. We do not discuss in detail what determines the high species diversity because this is a large and complex literature, and space is limited. However, we now include some limited discussion about which mechanisms may drive the consistent patterns of commonness we find, at line 333, noting that distance- or density-dependent juvenile mortality may both explain the maintenance of diversity, and the hyperdominance patterns (if these species are the ones that largely escape distance- or density-dependent mortality).

4. Also, rare species are extremely important for conserving the greatest biodiversity on the planet. It is important to clarify that it would be a bad strategy only to concentrate efforts on common species, neglecting rare species. It is also necessary to carry out research efforts and, above all, the conservation of rare species. Perhaps the concept of umbrella species can be used, indicating that conserving common species could also conserve rare species.

DLMC&SLL: We absolutely agree on conserving diversity – and now make the point more clearly, in the abstract and in the final sentence of the discussion that the focus on common species is to help us understand some aspects of tropical forest ecology, but should not be a guide to conservation. In the abstract we now say:

Our results should not detract from the importance of understanding and protecting rare species but do open new opportunities to understand the world's most diverse forests, including modelling their response to environmental change, by focusing on the common species that comprise the majority of their trees.

We finish the paper by saying (lines 384-391):

Of course, striving to understand and protect rare and non-hyperdominant species remains crucial, particularly as they face greater extinction risk and may also contribute to the functioning of ecosystems, particularly when more functions^{55,56}, longer timescales⁵⁷, and imposed environmental changes⁵⁶ are considered, and given that the hyperdominants of the future may be rarer today. Nonetheless, with a complementary grasp of the most common species, mapping, understanding, and modelling of the world's tropical forests will be a much more tractable proposition.

5. One aspect I suggest discussing is that the common species by abundance may not be the dominant species by biomass. Therefore, estimating the number and percentage of species that cover 50% of the biomass would be relevant. I can imagine that many common species are those that inhabit the understory and that they are not necessarily the ones that contribute a significant fraction of the forest's biomass. Common species by abundance and dominant species by biomass could play different roles in tropical forest ecosystems' structure, dynamics, and function.

DLMC&SLL: We agree and have added a sentence on this towards the end of the discussion. At line 375-382 we write:

In the future, our analyses should be extended to investigate forest carbon stocks and hyperdominant species and their role in the provision of ecosystem services. In Amazonia, even fewer tree species were found to account for 50% of aboveground carbon stocks than the minimum number required to account for 50% of trees¹⁶. Following these results, our expectation is that for carbon stock hyperdominance, common canopy tree hyperdominants will be carbon stock hyperdominants while some common understory species may not be carbon stock hyperdominants and may be replaced by a smaller number of large-stature species.

6. The authors indicate that a similar percentage of common species was found in the forests of the three continents. This implies the existence of a fundamental and general assembling rule of tree communities. However, the paper does not discuss what mechanisms may be underlying this rule. In the literature, different deterministic and neutral mechanisms have been proposed that could be used to expose ideas that help to understand if this is a statistical rule (for example, it has been found that ordering the words of a book by their frequency of appearance can generate a log-normal distribution) or it has tractable ecological hypotheses to explain the similarity in the low percentage of common species.

DLMC&SLL: We initially left this discussion out and agree it should be included, as also suggested by Rev#1 and #3. We now address this in the discussion at lines 312-345:

Nevertheless, consistent patterns of commonness do not necessarily imply the same causal mechanism(s). The ubiquity of the broad 'rare is common, common is rare' pattern in ecology, which is also found in non-biological complex systems⁴⁵, means inferences as to the cause of this broad pattern are challenging^{24,46}. While combinatoric methods⁴⁶ and models that maximise the entropy of information^{47,48} both produce the ubiquitous 'reverse lazy-J' pattern, empirical observations show fewer common species and more rare species than expected by statistical controls alone⁴⁶. Similarly, neutral models produce the same broad pattern, but produce too few individuals of the most

common Amazonian tree species⁴⁹. This suggests that biological mechanisms influence tree community assembly to produce a consistent proportion of common species across continents.

Considering the substantially smaller regional species pool in Africa compared to Amazonia and Southeast Asia, one might expect differing continental patterns of species dominance if evolutionary drivers were the primary mechanism, not the consistent patterns that we find. Similarly, if environmental filtering were a key mechanism, the different contemporary environments, with Africa much drier on average than the other two continents²², and Southeast Asia consisting of scattered island-like areas of forest compared to the contiguous forested region of Amazonia, would also imply differing continental patterns of species dominance, not the near-identical patterns that we find. These constraints limit the potential mechanisms that could apply across our three-continent context.

One potential cross-continental mechanism is dispersal limitation, where the dispersal capabilities of species result in some suitable habitat patches remaining unoccupied. Another mechanism is density- or distance-dependent mortality, which appears widespread across tropical forests⁵⁰. Here, specialist natural enemies such as pathogens and herbivores reduce seed or juvenile conspecific survival rates near conspecific adults or areas of high juvenile conspecific density, thereby reducing competitive exclusion and contributing to the maintenance of high tree species richness in tropical forests⁵⁰. It is possible that common species have largely evaded density- and/or distance-dependent mortality. Further progress on putative mechanisms can be made, for example, by exploring whether ecological or functional traits differ between common and rare species, and assessing the consistency of any differences among tropical continents⁵¹. While deducing mechanisms is complex, the identification of a tractable number of common species in tropical forests will facilitate progress in understanding of tropical forests beyond species-abundance distributions.

7. To go beyond the finding of the low percentage of common species prevalent throughout the planet's tropical forests, the authors could discuss (perhaps supported by data) the role that common species can play in the maintenance of forest biodiversity. For example, one could find out the frequency of common species pollinated by animals and the frequency of common species producing fleshy fruits eaten by frugivorous and seed-dispersing animals. How common species contribute to the structure of tree communities can also be discussed, for example, by accounting for the frequency of species that are in the understory and those that occupy the canopy. Finally, it would be important to highlight the role of common species as a source of ecosystem services. For example, in addition to functioning as carbon sinks and storage, common species can be a source of current or potential forest products for societies. This issue could be discussed shortly.

DLMC&SLL: We are not sure how to discuss the contribution of common species to the maintenance of diversity without being excessively speculative, so we have not added this. New analyses combining our approach with trait data is non-trivial so while we do suggest this as a fruitful avenue to follow (in line 272-280) we do not undertake it in this paper. We expect that our published list of common species will spur targeted new trait data collection and conduct these types of analyses in the future. The point about the role of common species for ecosystem services is well taken and we now add at line 375-382 in the discussion a section on carbon:

In the future, our analyses should be extended to investigate forest carbon stocks and hyperdominant species and their role in the provision of ecosystem services. In Amazonia, even fewer tree species were found to account for 50% of aboveground carbon stocks than the minimum number required to account for 50% of trees¹⁶. Following these results, our expectation is that for carbon stock hyperdominance, common canopy tree hyperdominants will be carbon stock hyperdominants while some common understory species may not be carbon stock hyperdominants and may be replaced by a smaller number of large-stature species.

Technical aspects

It would be useful to provide data on climate (for example, variation in total annual precipitation between sites) and elevation above sea level of the study locations in the supplementary information. These data can help the reader to understand the similarities/differences between the forest types that were included in the study beyond the information indicating that tropical forests that fall into the "Tropical and Subtropical Moist Broad Leaf Forest" biome delimited by the ecoregions map of the World Wildlife Fund were included.

DLMC&SLL: Added annual precipitation, average temperature, and elevation from WorldClim at lines 481-491.

I believe that a brief discussion should be included on the precision/confidence of the method used to estimate the global number of trees in tropical forests, considering that in some cases, the estimation for some regions was based on data from a single plot. This discussion may be indicated in the supplementary information.

DLMC&SLL: This is a misunderstanding based on our too-brief description. The number of trees in a region is based on the Crowther et al. estimates (derived from 429,775 ground-based estimates of tree density), and not our plots. We now make this clearer in the methods, writing:

The total number of tropical trees ≥ 10 cm trunk diameter in each of our regions was then estimated by summing tree abundances in countries in which we have at least one sampled plot from the 'map of Global Tree Density' (Crowther et al⁶⁴) (derived from 429,775 ground-based estimates of tree density) and masking according to the 'Tropical and Subtropical Moist Broadleaf Forests' borders in ArcGIS v3.10.1⁶⁵. Thus, we estimate that there are ~92 billion, ~331 billion trees, and ~217 billion trees in our Africa, Amazonia, and Southeast Asia regions respectively, totalling 640 billion trees. Including abundance from countries in the 'Tropical and Subtropical Moist Broadleaf Forests' biome in which we have no sampled plots, we estimate ~799 billion total trees across all of Earth's moist tropical forests.

Referee #3 (Remarks to the Author):

General comments

The underlying idea of the manuscript is that we could understand the (dynamics of) tropical forests and their response to environmental change if only a small proportion of species would make up the majority of the larger tree individuals (i.e., "hyperdominant species"), for which it would be feasible to collect data on their natural history. However, the order of magnitude of hyperdominant species and their differences between tropical regions is not known.

The study compiles a large data set of small plots in tropical forests across Africa, Amazonia and Southeast Asia to assess the proportion of hyperdominant species in these three regions, to extrapolate them to the entire closed-canopy forests of the tropics, and to derive a list of species in their data set that are likely to be hyperdominant.

The results show a striking similarity in the proportion of hyperdominant species among the 3 regions and extrapolation suggests that a manageably small number of tree species (around 1000) may account for half of Earth's trees (with diameter > 10 cm) found in the closed-canopy tropical forest biome. On overall, the underlying idea and the results of the study are interesting and highly relevant for applied and theoretical ecologists, but will also appeal to a broader audience. The striking similarity in the patterns of species abundances in tropical forests among continents is also likely to stimulate theoretical progress in explain this pattern.

DLMC&SLL: Thank you for the very clear summary and positive feedback.

Extrapolating the number of species and their abundances from a relatively small number of plots to the entire Amazonian region or even the entire tropics is a complex and very much debated undertaking. As a consequence, a number of recent studies have

contributed to this issue by testing different species abundance distributions and methods to remove bias in the extrapolation. The manuscript rests on a recent method by ter Steege et al. (2020) that derived biased-corrected richness estimates for the Amazonian tree flora by recognizing that conspecific spatial aggregation at the sampling scale, which is a serious source of bias of parametric estimates of species richness, increases with rarity.

I understand the difficulties to derive such estimates, but also the need to put the available information together to make educated guesses. Nevertheless I feel a bit uneasy regarding some assumptions of the methods. Although ter Steege et al. (2020) addressed the issue of non-random species distributions, the potential effect of non-random sampling (i.e., aggregated location of the sampling plots) is not addressed. To assess the robustness of the results I recommend a simulation study where realistic large spatially-explicit tree maps (which provide also known species abundance distributions) are generated by a process based model and sampled with spatial placement of the sampling plots similar to the actual one. A dynamic model with spatially-explicit dispersal is required to produce realistic distance-dependent correlations in the species composition among plots that can lead to bias if plots are aggregated. Doing such an analysis for sufficiently large tree maps should be feasible using the software of Thompson et al. (2020: DOI: 10.1111/2041-210X.13451).

DLMC&SLL: We agree that this potential effect of non-random sampling should be addressed and appreciate the suggestion. We address this by using a resampling approach, as this contains fewer assumptions than the approach suggested and is consistent within the manuscript as we use this approach elsewhere in the manuscript. We define the plots from the largest dataset, Amazonia, into obvious clusters, and sample 1 plot from each cluster and then estimate the % hyperdominance, then repeat sampling 2 plots from each cluster, and so on, up to 10 plots from each cluster (where available). This is now Extended Data Figure 2, also displayed below. The spatial clustering of plots makes no significant difference to the results (it appears to have a small effect, if anything with more concentrated hyperdominance when we remove the clustering of plots, but is well within the uncertainty), see top right graph, purple is the full dataset, black is 1 plot from each cluster. The top left graph is all the 1, 2... 10 runs (without confidence intervals so we can see the points more easily). We also compute Fishers alpha, below. We write in the main text, lines 107-108:

This consistency is not affected by the aggregated spatial distribution of plots within each region (Extended Data Fig. 2)

Additionally, we write in the Methods (lines 497-502) and add an Extended Data Figure 2:

To quantify the effect of the spatial clustering of plots, we compared results from the full Amazonia data, as the largest dataset, to those from subsets of the Amazonia data in which 1,2,3,...,10 plots were sampled from each spatial cluster. We found that spatial clustering had a negligible and not

statistically significant effect on hyperdominant percentage and fitted values of Fisher's α (Extended Data Fig. 2). Therefore, we retain all plots for our analyses to maximise sample sizes.

Extended Data Figure 2: Impact of spatial clustering of plots on rarefaction curves of hyperdominant percentage (first row) and Fisher's Alpha (second row) in the Amazonia data. Purple points and confidence intervals represent the full data; black points and confidence intervals represent a subset of the data in which one plot is sampled from each spatial cluster of plots; other coloured points represent subsets of the data in which 2,3,4,...,10 plots (or the total number of plots in the cluster) are sampled from each spatial cluster of plots. Confidence intervals are derived via the standard deviation across iterations of subsamples taken with replacement at each sampling point. Note that although resampling for rarefaction was done by subsampling tree inventory plots, the curves are re-plotted with an x-axis of number of stems.

Specific comments

Unfortunately, the pdf did not contain line and page numbers, I therefore refer to concrete places in the manuscript using the pdf page numbers, but only to the paragraph level.

DLMC&SLL: Apologies for this oversight on our part, this has been rectified in the revised manuscript.

page 8p2

Be concrete, what are the “fundamental rules” or “fundamental processes”? Species abundance distributions have been intensely studied with theoretical models (e.g., neutral theory and extensions that provide mechanistic interpretations of the parameters of the LogSeries). This unspecified term is overused since it appears also at page 7p1, page 8p2, page 10p2, page 16p1, page 16p2, and page 17p1.

DLMC&SLL: We agree that these terms ‘fundamental rules/processes’ were overused, we have removed all but two instances of ‘fundament processes/mechanisms’. We have removed all instances of ‘rules’ (on reflection that they are not really rules). We now are as specific as we think the data will allow, discussing past work on theoretical models and deductions from the new results, adding three new paragraphs in the discussion to consider what we can infer from our results and be more concrete (also at the request of Rev#1 and #2 who made related points), at lines 312-345:

Nevertheless, consistent patterns of commonness do not necessarily imply the same causal mechanism(s). The ubiquity of the broad ‘rare is common, common is rare’ pattern in ecology, which is also found in non-biological complex systems⁴⁵, means inferences as to the cause of this broad pattern are challenging^{24,46}. While combinatoric methods⁴⁶ and models that maximise the entropy of information^{47,48} both produce the ubiquitous ‘reverse lazy-J’ pattern, empirical observations show fewer common species and more rare species than expected by statistical controls alone⁴⁶. Similarly, neutral models produce the same broad pattern, but produce too few individuals of the most common Amazonian tree species⁴⁹. This suggests that biological mechanisms influence tree community assembly to produce a consistent proportion of common species across continents.

Considering the substantially smaller regional species pool in Africa compared to Amazonia and Southeast Asia, one might expect differing continental patterns of species dominance if evolutionary drivers were the primary mechanism, not the consistent patterns that we find. Similarly, if environmental filtering were a key mechanism, the different contemporary environments, with Africa much drier on average than the other two continents²², and Southeast Asia consisting of scattered island-like areas of forest compared to the contiguous forested region of Amazonia, would also imply differing continental patterns of species dominance, not the near-identical patterns that we find. These constraints limit the potential mechanisms that could apply across our three-continent context.

One potential cross-continental mechanism is dispersal limitation, where the dispersal capabilities of species result in some suitable habitat patches remaining unoccupied. Another mechanism is density- or distance-dependent mortality, which appears widespread across tropical forests⁵⁰. Here, specialist natural enemies such as pathogens and herbivores reduce seed or juvenile conspecific survival rates near conspecific adults or areas of high juvenile conspecific density, thereby reducing competitive exclusion and contributing to the maintenance of high tree species richness in tropical forests⁵⁰. It is possible that common species have largely evaded density- and/or distance-dependent mortality. Further progress on putative mechanisms can be made, for example, by exploring whether ecological or functional traits differ between common and rare species, and assessing the consistency of any differences among tropical continents⁵¹. While deducing mechanisms is complex, the identification of a tractable number of common species in tropical forests will facilitate progress in understanding of tropical forests beyond species-abundance distributions.

page 8p2, page 17p2

This is true, however, non-hyperdominant species may comprise a reservoir of species that become more important (dominant) and function as a buffer if environmental conditions change. See also the diversity-functioning debate. I think this should be discussed.

DLMC&SLL: We agree. We now write at the end of the paper, at line 384-391:

Of course, striving to understand and protect rare and non-hyperdominant species remains crucial, particularly as they face greater extinction risk and may also contribute to the functioning of ecosystems, particularly when more functions^{55,56}, longer timescales⁵⁷, and imposed environmental changes⁵⁶ are considered, and given that the hyperdominants of the future may be rarer today. Nonetheless, with a complementary grasp of the most common species, mapping, understanding, and modelling of the world's tropical forests will be a much more tractable proposition.

page 9p2

How does "trunk diameter" relate to dbh?

DLMC&SLL: dbh stands for diameter at breast height, and is measured 1.3 m along a tree stem or above any buttresses or deformities, We now clarify this at first use in line 81:

We limit our analysis to trees ≥ 10 cm trunk diameter at breast height (1.3m along the stem or above any buttresses or deformities), the widely used minimum size for inventoring tropical trees.

page 9p3

I miss an explanation why including smaller plots inflates the total number of species per

number of stems (Extended Data Fig. 1). I guess that this is an effect of species aggregation and beta diversity (distance decay of similarity) where two or more smaller but spatially separated plots host more species than one larger plot of their joined size. How does the remaining differences in plot sizes influence results? The simulation tests (e.g., ter Steege et al. 2020) of the method used 1-ha plots, were most sampling plots 1-ha plots?

DLMC&SLL: On smaller plots, we add the explanation in parentheses in the first sentence below. We have also added more information on the plot sizes in the methods, as shown below. In the main analysis in the manuscript we used the same dataset as the ter Steege et al. 2020 simulations, which are indeed mostly 1 ha but we include only plots >0.9 ha (the ATDN limit is just 0.1 ha, but there are very few of these smaller plots in the dataset). After imposing the 0.9 ha limit, all continents have a median plot size of 1 ha. This ensures the methods of Africa and Asia (where there are more small plots) are consistent with the Amazon plots and the previous work of ter Steege et al. Specifically, in our main analysis, in Africa and Amazonia most plots were 1 ha, at 88% and 90% of respective total number of plots, but only 48% plots were 1 ha in the Asia dataset. In response to this comment we have additionally re-run the rarefaction analysis using only 1 ha plots from Africa and S. America, and consistent near-identical percentage hyperdominance was found across these two continents. We now write in the methods, at line 475-487:

We found that small plots (<< 1 ha) inflate per-plot species totals relative to larger plots (because the rate of encountering new species is higher the smaller the plot size; Extended Data Fig. 1), so we limited our analyses to plots > 0.9 ha to enable like-for-like comparison. For Africa, we retained 368 plots covering 450 ha (mean plot area 1.22 ha, median 1 ha, range 0.92 – 10 ha; 2% of plots 0.9-0.99 ha, 88% 1 ha, 8% 1.01-5 ha, 1% >5ha). For Amazonia we retained 1,097 plots covering 1,434 ha (mean plot area 1.31 ha, median 1 ha, range 0.9 – 78.8 ha; 2% of plots 0.9-0.99 ha, 90% 1 ha, 7% 1.01-5 ha, 1% >5ha). For Southeast Asia we retained 103 plots covering 164 ha (mean plot area 1.59 ha, median 1 ha, range 0.96 – 4.5 ha; 1% of plots 0.9-0.99 ha, 48% 1 ha, 52% 1.01-5 ha, 0% >5ha). We assessed if the remaining differences in plot size affected the results, using only the 1 ha plots from Africa (n=323) and Amazonia (n=988), rarefied to the size of the Asia dataset, again finding near-identical percent hyperdominance on the two continents (Africa: 7.30%, 95% CI: 6.56-8.04; Amazonia: 7.35%, 95% CI: 6.61-8.10).

Given the sensitivity of results to this spatial issue, it is likely that the spatial arrangement of the plots, which is highly non-random, interacts with species aggregation and beta diversity in biasing the results. I therefore recommend that the authors test their analysis against realistic simulated spatially-explicit tree maps, e.g., using the software of Thompson et al. (2020).

DLMC&SLL: The sensitivity appears to be to the grain size (as we re-sample by plot), rather than space per se, as our analysis on plot clustering, addressed above, with new ED Fig. 2, shows no significant impact of this.

Page 10p1+2

Indeed, the pan-tropical consistency in the proportion of hyperdominant species is a striking result!

DLMC&SLL: We agree!

Page 12p1

Extended data Fig. 3 was not provided, I guess you mean Extended Data Figure 4? Should you here not also provide the estimated number of trees in each study region (that are later given at page 19)? The estimate of 299 hyperdominant species in Amazonia match the estimate of Tovo et al. (2017).

DLMC&SLL: Apologies – yes there was no ED Fig. 3, we meant ED Fig. 4. These figures have now been renumbered. We give the number of trees in each study region, now at lines 167, 169, and 170. On Tovo et al. 2017, we had not noticed that, thanks! We now cite Tovo et al. 2017, writing at line 172:

Our results from Amazonia match those derived using a different extrapolation approach³¹.

Page 15p1

Using simulated tree maps could also help to test the methodology to detect hyperdominant species.

DLMC&SLL: This is addressed in a previous response to this idea, and see ED Fig. 2.

Page 16p1+2

Here you should go deeper and discuss theories that may explain the fundamental rules of tree community assembly that may lead to consistent patterns of species abundance.

DLMC&SLL: We agree, please refer to our new paragraphs, noted in response to the comment on the same subject above.

Page 18p2+3+4

You should provide more detailed information on the size distribution of the plots. Tests of your methodology rest on 1-ha plots.

DLMC&SLL: This information is added. The 1 ha test is reported above.

Page 20p1

The spatial pattern of plots is also likely to introduce some bias into your estimates.

DLMC&SLL: Dealt with above, see new ED Figure 2.

Page 21p3

Please mention that the approach considered species-specific aggregation at the plot scale depending on species density.

DLMC&SLL: We have now added this at line 542-543:

Thus, this approach corrects for species-specific aggregation at the plot scale depending on species density.

Page 23p2

You mean Extended Data Fig. 5?

DLMC&SLL: Yes, apologies. ED Figures now renumbered!

* Nature Portfolio's authors website (<https://www.nature.com/authors>) contains information about and links to policies and resources.

This email has been sent through the Springer Nature Manuscript Tracking System NY-610A-SN&MTS

Confidentiality Statement:

This e-mail is confidential and subject to copyright. Any unauthorised use or disclosure of its contents is prohibited. If you have received this email in error please notify our Manuscript Tracking System Helpdesk team at <http://platformsupport.nature.com>.

Details of the confidentiality and pre-publicity policy may be found here <http://www.nature.com/authors/policies/confidentiality.html>

Privacy Policy | Update Profile

DISCLAIMER: This e-mail is confidential and should not be used by anyone who is not the original intended recipient. If you have received this e-mail in error please inform the sender and delete it from your mailbox or any other storage mechanism. Springer Nature Limited does not accept liability for any statements made which are clearly the sender's own and not expressly made on behalf of Springer Nature Ltd or one of their agents. Please note that Springer Nature Limited and their agents and affiliates do not accept any responsibility for viruses or malware that may be contained in this e-mail or its attachments and it is your responsibility to scan the e-mail and attachments (if any).

Springer Nature Ltd. Registered office: The Campus, 4 Crinan Street, London, N1 9XW. Registered Number: 00785998 England.

Reviewer Reports on the First Revision:

Referees' comments:

Referee #1 (Remarks to the Author):

I previously reviewed this work earlier in the year (Reviewer 1). My somewhat cynical take was that the results are demonstrating one of the few rules we have in ecology - few species are common and many are rare. I must admit I was a little surprised others found this to be a new insight and I think the authors seem to now make it clear that this isn't all that surprising. I am happy the authors have now included some citations of some key literature on this topic. I think that improves the manuscript and provides some needed context.

My argument that ID'ing mechanism from these patterns is known to be challenging seems to be accepted by the authors and they state this is where future work could focus. Again, at least one other reviewer had a totally different perspective.

I think by the end what the authors are pointing to as the big advance here is a list of common species that we could use to focus various types of ecological investigations. I find that useful and agree that that is something new. I'm not convinced is a major advance, however.

There are potentially interesting things to do with these types of data. As mentioned in my previous review, Ricklefs and Renner have investigated the phylogenetic/taxonomic composition of common and rare species in their Science paper (still not cited presently) in an attempt to assign some kind of deterministic mechanism. Webb and Pitman (2002) Sys Biol attempted to look for phylogenetic patterns in abundance distributions across two tropical floras. Finally, of course, Gentry in his Ann. Miss. Bot. Gard. papers notes consistencies in the taxonomic composition of dominant species across tropical forests worldwide. While these are somewhat smaller datasets, these works are useful to consider.

At the end, I'm afraid my perspective on this work hasn't changed too much. The authors now concede that the pattern isn't all that surprising and no mechanism can be identified. We all agree the list of species is new information.

I don't think my perspective would change that much with a revision, but if one were invited I would suggest the authors extend the analysis to temperate forests. There are many available datasets that could be used. If similar "hyper dominance" is uncovered would we be surprised? Is it worthwhile to have a list of common temperate trees? I think if these issues were considered it might help shine some light on whether the results from this tropical dataset are really a major breakthrough.

Referee #2 (Remarks to the Author):

I enjoyed reviewing the revised version of Cooper et al.'s paper. I found that most of the comments provided by the reviewers were addressed satisfactorily, particularly those I had suggested. I believe the abstract covers the key findings, and the paper is both original and of high scientific and practical interest. It is well-written, methodologically robust, with comprehensive data and analysis, and presents clear and relevant conclusions.

I have only one observation regarding the potential causes of hyperdominance in tropical forests. Recent studies have argued that the dominance of a few species, many of them of utilitarian value, in Amazonian and other African and Asian tropical forest regions could be linked to historical and long-term human activity (see, for example, Roberts et al. 2017. *Nature Plants* 3:201793. doi: 10.1038/nplants.2017.93; Roberts . 2022. *Jungle*. Penguin Press). Increasing attention is being given to the effects that human management of tropical forests has had on their structure and composition (see, for example, Levis et al. 2012. *PLoS ONE* 7:e48559. doi: 10.1371/journal.pone.0048559; Levis et al. 2018. *Frontiers in Ecology and Evolution*, doi: 10.3389/fevo.2017.00171). I suggest briefly mentioning this topic as one of the potential causes of hyperdominance in tropical forests.

Referee #3 (Remarks to the Author):

I am happy with the response to my comments and suggestions and glad that the plot sizes and the aggregation of plots does not affect the results. The resampling method that eliminated the aggregation of plots actually fits the overall methods of the manuscript better than my original suggestion

Overall I found that the manuscript has improved substantially, especially through addition of balanced discussion on the issues of underlying processes and research on abundant vs. rare species.

I congratulate the authors on their manuscript and the tremendous collaborative effort behind it.

Author Rebuttals to First Revision:

DLMC&SLL: We thank the reviewers of the paper for their feedback on the revised manuscript. All reviewers found that the revised manuscript improved substantially, and their positive feedback and constructive ideas are much appreciated.

The few remaining comments are addressed point-by-point below.

Referees' comments:

Referee #1 (Remarks to the Author):

I previously reviewed this work earlier in the year (Reviewer 1). My somewhat cynical take was that the results are demonstrating one of the few rules we have in ecology - few species are common and many are rare. I must admit I was a little surprised others found this to be a new insight and I think the authors seem to now make it clear that this isn't all that surprising. I am happy the authors have now included some citations of some key literature on this topic. I think that improves the manuscript and provides some needed context.

My argument that ID'ing mechanism from these patterns is known to be challenging seems to be accepted by the authors and they state this is where future work could focus. Again, at least one other reviewer had a totally different perspective.

I think by the end what the authors are pointing to as the big advance here is a list of common species that we could use to focus various types of ecological investigations. I find that useful and agree that that is something new. I'm not convinced is a major advance, however.

DLMC&SLL: As previously mentioned, that there are more rare species than common species is not interesting and is not surprising, but the fact that we find *almost identical* patterns of common species on different continents is interesting, and of importance. This quantification of patterns of common species has not been done for the tropics until now, which itself is an important advance. We appreciate that Reviewer 1 agrees that the identification of a list of common species is both useful and new.

There are potentially interesting things to do with these types of data. As mentioned in my previous review, Ricklefs and Renner have investigated the phylogenetic/taxonomic composition of common and rare species in their Science paper (still not cited presently) in an attempt to assign some kind of deterministic mechanism. Webb and Pitman (2002) Sys Biol attempted to look for phylogenetic patterns in abundance distributions across two tropical floras. Finally, of course, Gentry in his Ann. Miss. Bot. Gard. papers notes consistencies in the taxonomic composition of dominant species across tropical forests worldwide. While these are somewhat smaller datasets, these works are useful to consider.

DLMC&SLL: We agree that one next logical step is an investigation of the phylogenetic patterns in abundance distributions. But that is a different, and complex, paper which we hope to tackle in the future. We now cite Ricklefs and Renner (2012 Science), Webb and Pitman (2002) Sys Biol, and a newer paper, Silva de Miranda (2022, PNAS). We add to the discussion:

"Recent analyses have revealed that the same few families contribute most of the species richness in both Africa and Amazonia⁴⁹, which when combined with analyses showing that more diverse families have more common species⁵⁰, may indicate a role for deep evolutionary mechanisms driving the patterns we find."

At the end, I'm afraid my perspective on this work hasn't changed too much. The authors now concede that the pattern isn't all that surprising and no mechanism can be identified. We all agree the list of species is new information.

DLMC&SLL: The 'common is rare, rare is common' pattern is not surprising as we noted in the original manuscript, but the quantification on three continents of almost identical patterns of common species is both new and surprising to both the manuscript authors and the other two reviewers.

I don't think my perspective would change that much with a revision, but if one were invited I would suggest the authors extend the analysis to temperate forests. There are many available datasets that could be used. If similar "hyper dominance" is uncovered would we be surprised? Is it worthwhile to have a list of common temperate trees? I think if these issues were considered it might help shine some light on whether the results from this tropical dataset are really a major breakthrough.

DLMC&SLL: The hyper-diversity of tree species in the tropics means it is non-obvious which species are common species until large samples are obtained. Such data is difficult to collect when a single hectare of forest in Amazonia can have as many tree species as all of Western Europe combined. We do not think an analysis of temperate trees is warranted, as the identities of the small number of common tree species are already known in temperate North America and Eurasia.

Referee #2 (Remarks to the Author):

I enjoyed reviewing the revised version of Cooper et al.'s paper. I found that most of the comments provided by the reviewers were addressed satisfactorily, particularly those I had suggested. I believe the abstract covers the key findings, and the paper is both original and of high scientific and practical interest. It is well-written, methodologically robust, with comprehensive data and analysis, and presents clear and relevant conclusions.

DLMC&SLL: Thank you very much for the positive feedback!

I have only one observation regarding the potential causes of hyperdominance in tropical forests. Recent studies have argued that the dominance of a few species, many of them of utilitarian value, in Amazonian and other African and Asian tropical forest regions could be linked to historical and long-term human activity (see, for example, Roberts et al. 2017. *Nature Plants* 3:201793. doi: 10.1038/nplants.2017.93; Roberts . 2022. *Jungle*. Penguin Press). Increasing attention is being given to the effects that human management of tropical forests has had on their structure and composition (see, for example, Levis et al. 2012. *PLoS ONE* 7:e48559. doi: 10.1371/journal.pone.0048559; Levis et al. 2018. *Frontiers in Ecology and Evolution*, doi: 10.3389/fevo.2017.00171). I suggest briefly mentioning this topic as one of the potential causes of hyperdominance in tropical forests.

DLMC&SLL: We mention this theme in the following section (lines 289 – 299):

We find common diversity patterns despite the very different histories of human occupancy in Amazonia, African, and Southeast Asian tropical forests⁴¹. The relatively recent arrival of humans in Amazonia approximately 20,000 years ago has been linked to the greater Pleistocene extinctions, in contrast to much longer human occupancy in the tropical forests of Africa and Southeast Asia⁴². Some have also suggested that Amazonian forest composition was altered by humans through the incipient domestication of tree species, increasing the abundance of a small number of favoured species⁴³. Others have reported large areas of deforestation associated with the African Iron Age⁴⁴. How can such different human histories result in near-identical patterns of tree species dominance? The most parsimonious explanation is that the system tends to return to a state with a similar species abundance pattern.

Referee #3 (Remarks to the Author):

I am happy with the response to my comments and suggestions and glad that the plot sizes and the aggregation of plots does not affect the results. The resampling method that eliminated the aggregation of plots actually fits the overall methods of the manuscript better than my original suggestion

Overall I found that the manuscript has improved substantially, especially through addition of balanced discussion on the issues of underlying processes and research on abundant vs. rare species.

I congratulate the authors on their manuscript and the tremendous collaborative effort behind it.

DLMC&SLL: Thank you very much for the positive feedback!